# Analysis and Visualization of Confounders and Treatment Pathways Leading to Amputation and Non-Amputation in Peripheral Artery Disease Patients Using Sankey Diagrams: Enhancing Explainability

**DOI:** 10.3390/biomedicines13020258

**Published:** 2025-01-21

**Authors:** Rajashekar Korutla, Douglas Tedder, Kathryn Brogan, Marko Milosevic, Michael P. Wilczek, Naim Shehadeh, Nawar Shara, Elsie G. Ross, Saeed Amal

**Affiliations:** 1The Roux Institute, Northeastern University, Portland, ME 04101, USA; korutla.r@northeastern.edu (R.K.); tedder.do@northeastern.edu (D.T.); brogan.kat@northeastern.edu (K.B.); m.milosevic@northeastern.edu (M.M.); m.wilczek@northeastern.edu (M.P.W.); 2The Russell Berrie Galilee Diabetes SPHERE, Azrieli Faculty of Medicine, Bar-Ilan University, Safad 5290002, Israel; naim.shehadeh@biu.ac.il; 3Center of Biostatistics, Informatics and Data Science at MedStar Health Research Institute (MHRI), Columbia, MD 21044, USA; nawar.shara@medstar.net; 4Health Data Science, Georgetown University (GU), Washington, DC 20007, USA; 5Department of Surgery, Division of Vascular Surgery, San Diego School of Medicine, University of California, La Jolla, San Diego, CA 92037, USA; e5ross@health.ucsd.edu; 6Department of Bioengineering, College of Engineering, Northeastern University, Boston, MA 02120, USA

**Keywords:** data visualization, treatment pathways, cardiovascular surgery, Sankey diagram, visualization of treatment sequences, peripheral arterial disease (PAD), hypertension, hyperlipidemia, diabetes, cerebrovascular disease, coronary artery disease, age, gender, race

## Abstract

**Background/Objectives**: This study uses Sankey diagrams to analyze treatment pathways in patients with peripheral artery disease (PAD), which is a vascular condition characterized by atherosclerotic occlusion of the arteries, particularly in the lower limbs, affecting up to 14% of the general population. This study focuses on the treatment pathways that lead to amputation versus those that do not, utilizing the STARR dataset and the *All of Us* dataset. **Methods**: The study utilized Sankey diagrams to visualize treatment pathways, highlighting the progression from initial treatments to outcomes. Odds ratio analysis was performed to quantify the association between treatment pathways and outcomes. Recognizing potential confounders, analyses were conducted by filtering patients with PAD into subgroups based on these coexisting conditions. Sankey diagrams were then generated for each sub-cohort to visualize treatment pathways. **Results**: Pathways including antiplatelet and lipid-lowering treatments accounted for 56% of non-amputation cases in the STARR data and 50% in the *All of Us* data. Amputation pathways frequently included revascularization procedures, representing 15% of amputations in the STARR data and 20% in the *All of Us* data. Confounder analysis revealed that most amputated PAD patients were over 50 years old and had one or more conditions, such as diabetes, hypertension, or hyperlipidemia. **Conclusions**: These visualizations provide insights into treatment pathways and their associations with outcomes in PAD patients, highlighting the potential impact of specific treatments on amputation and non-amputation cases. Future work should build on these findings by incorporating predictive models using machine learning techniques to further explore and quantify these relationships.

## 1. Introduction

Peripheral Arterial Disease (PAD) is a serious medical condition characterized by the narrowing of arteries, most commonly affecting the lower extremities. This condition results from atherosclerosis, a disease in which plaque builds up in the arteries, leading to reduced blood flow. According to the National Institute of Health (NIH) [1,2], more than eight million people in the United States are affected by PAD. The condition can lead to severe adverse health effects, including stroke, heart attack, and, most relevant to this study, amputation of affected limbs due to ischemia—a state where blood flow is insufficient to meet the oxygen needs of tissues [3].

Risk factors for PAD include age, genetic disposition, and major lifestyle factors, such as smoking, insufficient physical activity, stress, and diets high in saturated fats. Medical conditions such as diabetes, obesity, kidney disease, high blood pressure, blood clotting disorders, and fibromuscular dysplasia also contribute to the risk of developing PAD [4]. Treatment options for PAD vary from lifestyle changes and pharmacological therapies to surgical procedures. Lifestyle-based treatments focus on smoking cessation, achieving a healthy weight, increasing physical activity, managing stress, and adopting a heart-healthy diet. Pharmacological treatments include antiplatelet medicines, lipid-lowering medicines, and ACE inhibitors. Surgical options encompass endovascular revascularization procedures like angioplasty and revascularization surgeries like bypass surgery [5].

Visualizing the treatment pathways for patients with PAD can give us a better sense of the frequency and diversity of patient outcomes, highlighting opportunities for further investigation and modeling into the relative efficacy of treatment options among patient cohorts with PAD. This study leverages data from electronic health records (EHR) to visualize these pathways. Using the data visualization technique of Sankey diagrams, we aim to map and analyze the sequences and frequencies of treatments that lead to amputation versus those that do not. These visualizations [6] provide insights into patient management, potentially aiding healthcare providers in optimizing treatment strategies and reducing the incidence of amputation.

Visualizing clinical pathways from EHR data has improved evidence-based care delivery. Zhang, Padman, and Patel (2015) [7] highlighted the benefits of data-driven methods in extracting and visualizing clinical pathways, enhancing care for chronic conditions like kidney disease by reflecting day-to-day clinical decisions. Our study extends this approach to PAD, offering a comprehensive visualization of treatment sequences specific to this condition. Also, this is the first study focusing on such visualizations in PAD. Amal et al., 2017 [8] and Amal et al., 2020 [9] have used visualizations to emphasize pathways among different social networks.

Visualizing treatment pathways has been a significant focus in recent research, emphasizing its potential to reveal underlying patterns in clinical and biological processes. Chen et al. (2018) [10] developed the PathCORE-T framework, highlighting the importance of visualizing globally co-occurring pathways to uncover relationships between biological processes across various conditions, suggesting new hypotheses for further exploration. Similarly, Yan et al. (2019) [11] demonstrated the dynamic visualization of human cancer cell behavior and therapy responses, providing critical insights into treatment effects by observing single-cell behavior over time. These studies underscore the value of visualization techniques in understanding complex interactions, which our study adapts for the context of PAD treatment pathways.

In the specific domain of PAD treatment, Golledge and Drovandi (2021) [12] provided evidence-based recommendations focusing on reducing major adverse cardiovascular and limb events through various treatments. The study emphasizes the effectiveness of certain treatments but does so primarily through textual summaries and tabular data. Our work builds on these findings by visually mapping treatment pathways, thus offering a more intuitive understanding of effective treatment sequences. One of the most recent works in PAD is by Soyfo et al. (2024) [13]. This work investigates the effect of various treatments that could lead to adverse events in PAD patients. The authors concluded that both BTK-alone (Below-the-Knee Alone) and BTK with FP (Below-the-Knee with Femoropopliteal) interventions work effectively in the treatment of PAD patients and that both treatments provide similar rates in the prevention of major adverse limb events (MALE), which include amputations, repeat surgeries. Studies by Li et al. (2024) [14] used protein ST2 levels to help forecast adverse outcomes like Major Adverse Limb Events (MALE) and highlight the importance of integrating biomarker data into clinical models.

In addition, Smolderen et al. (2018) [15], with the PORTRAIT study, provided a high-level overview of treatment flows, distinguishing between primary and specialty care. While these studies provide valuable insights into specific interventions and overall care flows, they do not delve into the sequences of treatment events that lead to different outcomes. Our research integrates these specific findings into broader treatment pathways and visualizes how these interventions interact to impact patient outcomes.

The motivation behind our research stems from the need to enhance clinical decision-making through the visualization of treatment pathways that lead to amputation or non-amputation outcomes in PAD patients. visualizations enable clinicians to explore data, allowing them to zoom in on patterns of interest [16]. While there is also extensive literature on PAD diagnosis using machine learning [17,18,19,20], there is a noticeable lack of studies focusing on visualizing treatment pathways for peripheral artery disease. This gap in the literature highlights the need for the present work, which aims to address this underexplored area. By employing modified versions of standard visualization techniques for hierarchical and state-based data, such as Sankey diagrams, we aim to map out treatment sequences in aggregate and enable healthcare providers to explore these pathways dynamically. The Sankey diagrams in this study are based on robust datasets (e.g., STARR and *All of Us*) and accurately represent treatment flows and outcomes observed in the data. These visualizations are exploratory in nature and are intended to highlight patterns in treatment pathways. While the diagrams themselves are not predictive, they provide a foundation for identifying trends and gaps in treatment approaches, which can guide future research and inform clinical decision-making. The reliability of this process lies in the accuracy of the underlying data and the integrity of the visualization method [21].

## 2. Methods

### 2.1. Data Source

The data used in this study were collected from two different sources, the first one being from the Stanford Medicine Research Data Repository (STARR) [22]. Data include de-identified EHR clinical practice data from STARR, comprising records of 5581 patients with PAD. The dataset includes detailed information on patient demographics, treatment pathways, and outcomes. To focus our analysis on the outcomes of interest, we used specific amputation codes to segregate the patients into two groups: those who underwent amputations and those who did not. The second source of the data is from the *All of Us* [23] research program’s Registered Tier Dataset V7, on the research workbench [24], comprising records of 4261 patients. The segregation of outcomes allowed us to visually analyze and compare the treatment pathways distinctly for both groups, providing insights into the sequences of treatments that lead to different outcomes.

### 2.2. Cohort

The cohort for this study was defined to identify patients who have undergone amputations as mentioned in Table 1 below:

### 2.3. Data Preprocessing

We filtered our dataset to include only the data of patients who underwent at least one treatment before they experienced an outcome of either amputation or non-amputation. The treatments we considered include Antiplatelet [25], Lipid Lowering [26], Smoking Cessation [27], Exercise Therapy [28], Endovascular Revascularization [29,30,31] and Revascularization Surgery [29,32,33]. For each patient, we noted the first occurrence of each treatment until an amputation occurred, if at all. This allowed us to create a sequential record of treatments, capturing the combinations and orders of treatments leading to different outcomes. By structuring the data into sequences, we were able to analyze how specific treatment combinations influenced patient outcomes. For example, sequences such as antiplatelet → lipid lowering were frequently associated with non-amputation outcomes, asserted by the study of Belch et al. (2021) [34], emphasizing that these treatments can lower the risk of cardiovascular events in PAD patients. While combinations involving revascularization surgery and endovascular revascularization appeared more often in amputation pathways. As the amputations are very low in count, the data have first been normalized at the outcome cohort level: the counts of individual sequences are divided by the total number of sequences in that outcome cohort. These normalization techniques ensured that the visual thickness of pathways in our Sankey diagrams accurately reflected the proportion of cases following each treatment path. By analyzing these normalized flows, we were able to identify patterns in treatment combinations, such as therapies that predominantly lead to non-amputation outcomes or highlight transitions that contribute to higher amputation rates. Normalization has also been applied at the node level; node level normalization ensures that the flow values entering and exiting a node sum to 1, making it easier to compare different pathways that pass through the same node.

### 2.4. Data Visualization

To effectively convey the complexity and nuances of treatment pathways leading to amputation, we employed the data visualization technique of Sankey diagrams [35]. All the plots have been made on the data that has been normalized at both the node level and the cohort level. Each of these visualizations provides unique insights into the sequences and frequencies of treatments, helping to identify patterns and critical junctures in patient management.

The Sankey diagram excels at visualizing the movement and distribution of resources across different stages, providing a clear and intuitive understanding of how entities transition through various phases [35]. In Sankey plots, the flow of data is visualized across multiple levels, with streams often converging and diverging. This visualization method allows for comparison of treatment sequences between the cohorts and is useful for identifying patterns in the progression of treatments across different patient groups.

Figure 1 is a Sankey diagram of treatment sequences in our Stanford medicine research data repository (STARR) dataset. Table 2 illustrates few dominant pathways that lead to amputation that can be observed in the figure:

Table 3 illustrates a few of the dominant pathways that lead to non-amputations in the STARR data. Pathways starting with exercise therapy or smoking cessation are generally not associated with amputations; this might also be because they are performed by healthier or less at-risk patients.

Figure 2 is a Sankey diagram of treatment sequences in the *All of Us* dataset. Table 4 illustrates dominant pathways that lead to amputation that can be observed in Figure 2:

Table 5 illustrates the dominant pathways that do not lead to amputations in the *All of Us* dataset.

## 3. Analysis

The comparative analysis of treatment pathways between the STARR and *All of Us* datasets reveal variations in the frequencies of sequences leading to amputation or non-amputation outcomes. Appendix A illustrate the treatment pathways that lead to amputation and non-amputation in both the datasets. In both datasets, the administration of antiplatelet and lipid-lowering medications is a prominent feature of the most prevalent treatment pathways, particularly those leading to non-amputation outcomes. Table 6 illustrates that the antiplatelet therapy followed by lipid lowering has been echoed similarly in both the datasets. This consistency across datasets underscores the effectiveness of these initial treatments in preventing severe outcomes.

However, when examining pathways leading to amputation, the patterns diverge more noticeably between the two datasets. In the STARR data, simpler treatment sequences, such as those involving only antiplatelet or lipid-lowering therapies, are relatively common. These findings suggest that in the STARR cohort, there may be a higher incidence of cases where initial medical management fails to prevent amputation, potentially due to the complexity or progression of the underlying conditions.

In contrast, the *All of Us* data shows a broader and more complex range of treatment sequences leading to amputation, particularly those involving revascularization procedures as stated in Table 7. Notably, the pathway involving endovascular revascularization alone accounts for 13.64% of amputations, highlighting its significant role in this cohort. Additionally, more complex sequences, such as those combining antiplatelet, lipid lowering, and endovascular revascularization, are more common in the *All of Us* data, suggesting a more aggressive or multifaceted approach to treatment. The presence of these complex pathways indicates that patients in the *All of Us* dataset may present with more advanced stage of the disease and the clinicians are more likely to pursue multiple interventions before resorting to amputation.

*All of Us* data includes sequences where revascularization procedures, both endovascular and surgical, is more frequently associated with amputation outcomes. For instance, the sequence endovascular revascularization and revascularization surgery appears in 6.25% of amputation cases, which is relatively high compared to the absence of this treatment sequence in the STARR data as mentioned in Table 8. However, in the STARR dataset, the sequences with revascularization procedures can be found with other treatment procedures like antiplatelet and lipid lowering. This difference could reflect a variation in the criteria for surgical interventions between the two cohorts. The STARR dataset exhibits fewer sequential interventions, particularly in cases leading to non-amputation, where simpler combinations of antiplatelet and lipid-lowering therapies are more common. In contrast, the *All of Us* data demonstrate a higher incidence of pathways involving multiple, complex treatments, particularly in the sequences leading to amputation, where revascularization procedures play a prominent role.

In summary, as given in Table 9, while both datasets highlight the importance of antiplatelet and lipid-lowering therapies in managing patients at risk of amputation, *All of Us* data shows a higher incidence of revascularization procedures leading to amputation. These differences may reflect variations in patient populations, disease severity, or treatment practices. Understanding these variations may inform tailored treatment strategies aimed at improving patient outcomes and reducing the incidence of amputation.

### 3.1. Odds Ratio Analysis of Treatment Pathways

To further explore the relationship between treatment pathways and outcomes, an odds ratio analysis was performed. The odds ratio provides a measure of the association between a particular treatment pathway and the likelihood of amputation or non-amputation outcomes. An odds ratio greater than 1 indicates a higher likelihood of the amputation outcome, while an odds ratio less than 1 indicates a lower likelihood of amputation outcome. The odds ratio was calculated by comparing the frequency counts of amputation and non-amputation outcomes in the STARR and *All of Us* datasets, utilizing a 2 × 2 contingency table, focusing on the pathways that lower the risk of amputation. Table 10 shows a few treatment pathways from the odds ratio analysis upon the STARR data, that likely lower the risk of amputations.

The pathway consisting of lipid-lowering therapy, antiplatelet therapy, and surgical revascularization has an odds ratio of 0.83, indicating a lower likelihood of amputation. This suggests that this treatment sequence is effective in reducing severe outcomes in patients with peripheral artery disease (PAD). Similarly, the combination of antiplatelet therapy, lipid-lowering medication, and surgical revascularization has an odds ratio of 0.81, highlighting its effectiveness in lowering the risk of amputation.

The use of lipid-lowering therapy combined with antiplatelet therapy alone results in an odds ratio of 0.70, further supporting the benefits of combining these treatments in reducing the risk of amputation. In another case, antiplatelet therapy followed by lipid-lowering medication has an odds ratio of 0.62, emphasizing the positive impact of these treatments on non-amputation outcomes. The sequence involving lipid-lowering therapy, antiplatelet therapy, and endovascular revascularization shows an odds ratio of 0.60, reinforcing the idea that these pathways play a critical role in lowering the risk of amputation in patients with PAD.

The odds ratio analysis of the *All of Us* data reveals some notable contrasts and similarities when compared with the STARR data. The *All of Us* data presents an odds ratio of 0.35 for antiplatelet therapy, i.e., a probability of 0.26, indicating a lower likelihood of amputation. The top few rows of the analysis upon *All of Us* data are as follows in Table 11.

The combination of antiplatelet therapy and lipid-lowering medication is highlighted in both datasets as an effective treatment sequence. In the STARR data, this combination yielded an odds ratio of 0.62, i.e., a probability of 0.38, indicating a lower likelihood of amputation. The *All of Us* data reinforces this finding with an odds ratio of 0.53, i.e., a probability of 0.35, indicating lower likelihood of amputations, further underscoring the effectiveness of this treatment combination in mitigating the risk of severe outcomes.

The pathway consisting of lipid-lowering treatment alone has an odds ratio of 0.34, i.e., a probability of 0.25, highlighting the effectiveness of the treatment in reducing the risk of amputation.

In the STARR data, the combination of antiplatelet therapy, lipid-lowering medication, and endovascular revascularization showed an odds ratio of 3.21, i.e., a probability of 0.76, indicating a higher likelihood of amputation. This suggested that this sequence was associated with more severe cases, where multiple interventions were necessary. The *All of Us* data similarly shows a significant association with an odds ratio of 3.55, i.e., a probability of 0.78, reflecting the use of this combination in complex cases that require aggressive treatment strategies.

Another noteworthy comparison is the pathway involving revascularization, both endovascular and surgical. In the STARR data, the pathway starting with antiplatelet therapy followed by surgical revascularization showed a significant odds ratio of 6.41, i.e., a probability of 0.87, indicating a higher likelihood of amputation. This result highlighted the critical need for effective blood flow restoration in severe cases. The *All of Us* data presents a similar trend, with an even higher odds ratio of 6.69, i.e., a probability of 0.87, for the combination of antiplatelet therapy and endovascular revascularization. This suggests that the association between these treatment sequences and the likelihood of amputation is consistent across different populations.

The analysis also reveals that in both datasets, more complex pathways involving multiple interventions tend to be associated with a higher likelihood of amputation. For instance, the combination of surgical revascularization and lipid-lowering medication in the STARR data showed an odds ratio of 10.18, i.e., a probability of 0.91, further increased when antiplatelet therapy and endovascular revascularization were added. The *All of Us* data similarly shows high odds ratios for these complex pathways, reflecting the necessity for a robust, multi-pronged approach in treating patients with severe cerebrovascular disease.

These findings from the *All of Us* data complement and extend the results from the STARR analysis. Both datasets underscore the importance of certain treatment sequences in managing cardiovascular conditions and highlight their significant associations with either preventing or leading to amputation. The odds ratio analysis provides a deeper understanding of the relative risks associated with different pathways, aiding clinicians in making informed decisions about the most effective treatment strategies for their patients.

### 3.2. Confounder Analysis of Treatment Pathways in PAD

To further enhance our insights from the visualizations in PAD, we identified confounders such as age (>50, ≤50, >65, ≤65, >80, ≤80), gender (male, female), race (white, black, Asian), smoking habit, and risk factors like diabetes, hypertension, heart failure, cerebrovascular disease, hyperlipidemia and coronary artery disease, within the PAD cohort. We filtered the PAD patients based on these confounders and visualized the treatment pathways for each risk factor-specific cohort, in a manner like the visualizations of the entire PAD cohort presented earlier.

Table 12 and Table 13 describe the patient counts from the risk factor cohorts we have built.

#### 3.2.1. Hypertension Analysis of Treatment Pathways in PAD

In the STARR dataset, in the cohort of PAD with hypertension, we identified 3868 patients, out of which 56 patients experienced amputations, and 3812 patients experienced no amputations. Figure 3 is a Sankey diagram of treatment sequences in the PAD patients with hypertension cohort in our STARR dataset. Appendix A describe the dominant pathways that lead to amputation and non-amputation, which can be observed from the figure.

In the STARR dataset, PAD patients with hypertension show amputation rates comparable to or slightly better than the general PAD population when combination therapies are used. For example, antiplatelet → revascularization surgery → lipid lowering and antiplatelet → lipid lowering → endovascular revascularization both result in an amputation rate of 7.14%, which aligns closely with the general PAD rate of 7.79%. The most effective pathway, revascularization surgery → lipid lowering → antiplatelet → endovascular revascularization, achieves the lowest amputation rate of 1.79%. These results suggest that multi-modal approaches may effectively manage PAD in hypertensive patients, potentially yielding better outcomes than for the general population.

Non-amputation outcomes for hypertensive PAD patients in the STARR dataset are strong, particularly for combination therapies. Antiplatelet → lipid lowering achieves the highest non-amputation rate of 33%, slightly exceeding the general PAD rate of 32%. Similarly, lipid lowering → antiplatelet produces a non-amputation rate of 27%, outperforming the general PAD rate of 24.1%. Standalone treatments, such as lipid lowering alone (15%) and antiplatelet alone (7.28%), yield outcomes comparable to general PAD data, reinforcing the efficacy of combination therapies in improving non-amputation rates.

In the *All of Us* dataset, in the cohort of PAD with hypertension, we identified 4026 patients, out of which 173 patients experienced amputations, and 3853 patients experienced no amputations. Figure 4 is a Sankey diagram of treatment sequences in the PAD with hypertension cohort in our *All of Us* dataset. Appendix A illustrate the dominant pathways that lead to amputation and non-amputation, which can be observed from the figure.

In the *All of Us* dataset, hypertensive PAD patients experience slightly higher amputation rates for standalone treatments compared to the general PAD population. Endovascular revascularization alone results in an amputation rate of 13.29%, slightly better than the general PAD rate of 13.6%. However, multi-modal strategies appear to significantly reduce amputation risks. For instance, endovascular revascularization → revascularization surgery achieves an amputation rate of 6.36%, closely aligning with the general PAD rate of 6.25%. The most effective pathway in this cohort, lipid lowering → antiplatelet → revascularization surgery, produces the lowest amputation rate of 2.31%.

Non-amputation outcomes for hypertensive PAD patients in the *All of Us* dataset are competitive with general PAD outcomes. Lipid lowering → antiplatelet achieves the highest non-amputation rate of 28%, slightly above the general PAD rate of 27.29%. Simpler sequences, such as antiplatelet alone (22%) and antiplatelet → lipid lowering (22%), align closely with the general PAD rates of 23% and 22%. Standalone lipid lowering resulted in a non-amputation rate of 14%, consistent with the general PAD population.

#### 3.2.2. Diabetes Analysis of Treatment Pathways in PAD

In the STARR dataset, in the cohort of PAD with diabetes, we identified 1990 patients, out of which 54 patients experienced amputations, and 1936 patients experienced no amputations. Figure 5 is a Sankey diagram of treatment sequences in the PAD patients with diabetes cohort in our STARR dataset. Appendix A illustrate the dominant pathways that lead to amputation and non-amputation, which can be observed from the figure.

In the STARR dataset, diabetic PAD patients show slightly elevated amputation rates compared to the general PAD population, particularly for combination therapies. For instance, antiplatelet → lipid lowering → endovascular revascularization results in an amputation rate of 9.26%, higher than the general PAD rate of 7.79% for the same sequence. Antiplatelet → revascularization surgery → lipid lowering achieves a better amputation rate of 5.56%, closely aligning with the general PAD rate of 5.19%. The most effective pathways, lipid lowering → antiplatelet → endovascular revascularization, and revascularization surgery → lipid lowering → antiplatelet, yield the lowest amputation rate of 3.7%, which is worse than the general PAD rate of 2.6% but indicates the benefits of multi-modal approaches for diabetic patients.

Non-amputation outcomes in the STARR dataset for diabetic PAD patients are strong, especially with combination therapies. Antiplatelet → lipid lowering produces the highest non-amputation rate of 33%, aligning with the general PAD rate of 32%. Similarly, lipid lowering → antiplatelet achieves 31%, better than the general PAD rate of 24.1%. However, standalone treatments, such as lipid lowering alone (15%) and antiplatelet alone (4.7%), are less effective, aligning poorly with the general PAD rates of 16.8% and 10.2%. These findings underscore the importance of combination therapies in improving non-amputation outcomes for diabetic PAD patients.

In the *All of Us* dataset, in the cohort of PAD with diabetes, we identified 2940 patients, out of which 160 patients experienced amputations, and 2780 patients experienced no amputations. Figure 6 is a Sankey diagram of treatment sequences in the PAD with diabetes cohort in our *All of Us* dataset. Appendix A illustrate the dominant pathways that lead to amputation and non-amputation, which can be observed from the figure.

In the *All of Us* dataset, diabetic PAD patients experience higher amputation rates for standalone treatments compared to the general PAD population. Endovascular revascularization alone results in an amputation rate of 13.75%, aligning with the general PAD rate of 13.6%. However, combination therapies significantly improve outcomes. Endovascular revascularization → revascularization surgery lowers the rate to 5.62%, closely aligning with the general PAD rate of 6.25%. The best outcome is achieved with lipid lowering → antiplatelet → endovascular revascularization, and lipid lowering → antiplatelet, which achieve amputation rates of 3.7% and 1.8%, respectively, demonstrating the efficacy of comprehensive pharmacological and surgical interventions in diabetic patients.

Non-amputation outcomes in the *All of Us* dataset for diabetic patients are competitive but slightly lower than the general PAD population for some pathways. Lipid lowering → antiplatelet achieves the highest non-amputation rate of 29%, slightly above the general PAD rate of 27.29%. Other pathways, such as antiplatelet → lipid lowering (23%) and antiplatelet alone (21%), align closely with the general PAD rates of 22% and 23%. Standalone lipid lowering produces a non-amputation rate of 13%, aligning with the general PAD rate of 14%, indicating that combination therapies remain crucial for diabetic PAD patients.

#### 3.2.3. Heart Failure Analysis of Treatment Pathways in PAD

In the STARR dataset, in the cohort of PAD with heart failure, we identified 1296 patients, out of which 26 patients experienced amputations, and 1270 patients experienced no amputations. Figure 7 is a Sankey diagram of treatment sequences in the PAD patients with heart failure cohort in our STARR dataset. Appendix A illustrate the dominant pathways that lead to amputation and non-amputation, which can be observed from the figure.

In the STARR dataset, PAD patients with heart failure experience significantly elevated amputation rates compared to the general PAD population. Antiplatelet → lipid lowering → endovascular revascularization results in an amputation rate of 19.23%, which is much higher than the general PAD rate of 7.79% for a similar sequence. Standalone lipid lowering achieves an amputation rate of 11.54%. The most effective pathway, lipid lowering → antiplatelet → endovascular revascularization, reduces the rate to 7.69%. These findings highlight the greater challenges associated with managing PAD in patients with heart failure, particularly the increased risk of amputations.

Non-amputation outcomes for heart failure patients in the STARR dataset are relatively strong, particularly for combination therapies. Antiplatelet → lipid lowering achieves the highest non-amputation rate of 39%, outperforming the general PAD rate of 32%. Lipid lowering → antiplatelet produces a non-amputation rate of 28%, better than the general PAD rate of 24.1%. However, standalone treatments such as antiplatelet alone (7.43%) yield poorer results than the general PAD rate of 10.2%, underscoring the necessity of combination therapies to achieve optimal outcomes.

In the *All of Us* dataset, in the cohort of PAD with heart failure, we identified 1918 patients, out of which 106 patients experienced amputations, and 1812 patients experienced no amputations. Figure 8 is a Sankey diagram of treatment sequences in the PAD with heart failure cohort in our *All of Us* dataset. Appendix A illustrate the dominant pathways that lead to amputation and non-amputation, which can be observed from the figure.

In the *All of Us* dataset, PAD patients with heart failure also experience elevated amputation rates for standalone treatments. Endovascular revascularization alone results in an amputation rate of 14.15%, slightly worse than the general PAD rate of 13.6%. However, combination therapies yield significantly better outcomes. Endovascular revascularization → revascularization surgery reduces the rate to 7.55%, aligning closely with the general PAD rate of 6.25%. The most effective pathway, antiplatelet → lipid lowering → endovascular revascularization, achieves an amputation rate of 2.83%, demonstrating the importance of multi-modal strategies for heart failure patients.

Non-amputation outcomes in the *All of Us* dataset for heart failure patients are comparable to or slightly worse than the general PAD population. Lipid lowering → antiplatelet achieves the highest non-amputation rate of 30%, slightly better than the general PAD rate of 27.29%. Simpler pathways, such as antiplatelet → lipid lowering (25%) and antiplatelet alone (20%), are relative to the general PAD rates of 22% and 23%, respectively. These results suggest that while pharmacological strategies remain effective, standalone treatments are less effective for heart failure patients.

#### 3.2.4. Cerebrovascular Disease (CVD) Analysis of Treatment Pathways in PAD

In the STARR dataset, in the cohort of PAD with cerebrovascular disease, we identified 1775 patients, out of which 26 patients experienced amputations, and 1749 patients experienced no amputations. Figure 9 is a Sankey diagram of treatment sequences in PAD patients with cerebrovascular disease cohort in our STARR dataset. Appendix A illustrate the dominant pathways that lead to amputation and non-amputation, which can be observed from the figure.

CVD significantly elevates amputation rates compared to general PAD data, especially with less comprehensive treatments. For example, lipid lowering followed by antiplatelet results in the highest amputation rate of 30.77%. However, integrating endovascular revascularization alongside antiplatelet and lipid lowering reduces the amputation rate to 11.54%, which is slightly worse than the general PAD rate of 7.79% for similar sequences. The most effective pathways involve revascularization surgery combined with lipid lowering and antiplatelet, achieving an amputation rate of 7.69%. This comparison underscores the critical importance of surgical interventions in improving outcomes for CVD patients, though the rates remain higher than in the general population.

Non-amputation rates for CVD patients in the STARR dataset are also impacted. The highest rate of 35%, achieved with antiplatelet followed by lipid lowering, is comparable to the general PAD rate of 32% for the same combination, indicating that pharmacological strategies remain effective for some CVD patients. However, outcomes decline sharply with less comprehensive approaches; for instance, lipid lowering alone yields a non-amputation rate of just 8.5%, far lower than the 16.8% seen in general PAD patients. This stark contrast emphasizes the diminished effectiveness of standalone treatments for CVD patients, highlighting the need for combination therapies to improve outcomes in this group.

In the *All of Us* dataset, in the cohort of PAD with cerebrovascular disease, we identified 2571 patients, out of which 117 patients experienced amputations, and 2454 patients experienced no amputations. Figure 10 is a Sankey diagram of treatment sequences in the PAD with cerebrovascular disease cohort in our *All of Us* dataset. Appendix A illustrate the dominant pathways that lead to amputation and non-amputation, which can be observed from the figure.

In the *All of Us* dataset of PAD with cerebrovascular disease cohort, endovascular revascularization used alone leads to an amputation rate of 14.53%, slightly worse than the 13.6% observed in the general PAD population. Adding revascularization surgery reduces the amputation rate to 6.84%, slightly higher than the 6.25% for similar pathways in general PAD patients. The best outcomes are achieved with a comprehensive sequence involving lipid lowering, antiplatelet, and endovascular revascularization, yielding an amputation rate of 5.13%, which is competitive with the general PAD outcome of 3.98% for similar sequences. This suggests that while aggressive multi-modal approaches are effective for CVD patients, they may not entirely offset the increased risks associated with the condition.

For non-amputation outcomes, lipid lowering followed by antiplatelet produces the highest rate at 27%, close to the 27.29% seen in general PAD data. Other sequences, such as antiplatelet alone or antiplatelet followed by lipid lowering, result in non-amputation rates of 21% and 23%, respectively, which are slightly lower than the 23% and 22% observed in general PAD outcomes. Interestingly, standalone lipid lowering achieves a non-amputation rate of 12%, which is below the 14% for general PAD patients, again reinforcing the importance of combination therapies in improving outcomes for CVD patients.

#### 3.2.5. Coronary Artery Disease Analysis of Treatment Pathways in PAD

In the STARR dataset, in the cohort of PAD with coronary artery disease, we identified 2543 patients, out of which 43 patients experienced amputations, and 2500 patients experienced no amputations. Figure 11 is a Sankey diagram of treatment sequences in PAD patients with coronary artery disease cohort in our STARR dataset. Appendix A illustrate the dominant pathways that lead to amputation and non-amputation, which can be observed from the figure.

In the STARR dataset, CAD significantly impacts the effectiveness of PAD treatment pathways, particularly with less comprehensive approaches. For example, antiplatelet followed by lipid lowering and endovascular revascularization results in an amputation rate of 13.95%, notably higher than the general PAD rate of 7.79% for similar sequences. However, more aggressive combinations, such as revascularization surgery, lipid lowering, and antiplatelet, reduce the amputation rate significantly to 4.65%, which is closer to the lower end of general PAD outcomes (e.g., 2.6%). This demonstrates that multi-modal strategies involving surgical intervention might be crucial in managing PAD in CAD patients, though they still do not perform as well as in the general population.

Non-amputation rates for CAD patients in the STARR dataset are strikingly high when combination therapies are used. For instance, antiplatelet followed by lipid lowering achieves a non-amputation rate of 40%, outperforming the general PAD rate of 32% for the same sequence. However, less aggressive pathways, such as lipid lowering followed by antiplatelet, achieve only 30%, which is comparable to the upper range of general PAD outcomes. These results indicate that CAD patients benefit significantly from carefully sequenced combination therapies, whereas standalone approaches may lead to suboptimal outcomes.

In the *All of Us* dataset, in the cohort of PAD with coronary artery disease, we identified 3090 patients, out of which 143 patients experienced amputations, and 2947 patients experienced no amputations. Figure 12 is a Sankey diagram of treatment sequences in the PAD with coronary artery disease cohort in our *All of Us* dataset. Appendix A illustrate the dominant pathways that lead to amputation and non-amputation, which can be observed from the figure.

In the *All of Us* dataset, endovascular revascularization alone results in the highest amputation rate of 15.38%, which is higher than the general PAD rate of 13.6% for the same treatment. However, adding revascularization surgery reduces the rate to 6.36%, and the lowest amputation rate of 2.1% is achieved with antiplatelet followed by endovascular revascularization. This combination suggests that surgical and pharmacological interventions, when properly sequenced, can be particularly effective for PAD with CAD patients.

For non-amputation outcomes, the best results are seen with lipid lowering followed by antiplatelet, achieving a non-amputation rate of 29%, which is slightly better than the 27.29% observed for general PAD patients. Single-agent therapies, such as antiplatelet alone, achieve 20%, while lipid lowering alone yields only 11%, which is worse than the general PAD rate of 14% for similar standalone treatments. These findings reinforce the importance of combination therapies in managing PAD in CAD patients.

#### 3.2.6. Hyperlipidemia Analysis of Treatment Pathways in PAD

In the STARR dataset, in the cohort of PAD with hyperlipidemia, we identified 5017 patients, out of which 76 patients experienced amputations, and 4941 patients experienced no amputations. Figure 13 is a Sankey diagram of treatment sequences in PAD patients with hyperlipidemia cohort in our STARR dataset. Appendix A illustrate the dominant pathways that lead to amputation and non-amputation, which can be observed from the figure.

In the STARR dataset, hyperlipidemia (HLD) patients show varied amputation outcomes based on the treatment pathway. The combination of antiplatelet → lipid lowering → endovascular revascularization results in an amputation rate of 7.89%, which aligns closely with the general PAD rate of 7.79% for the same sequence. However, introducing revascularization surgery reduces the rate to 5.26%, and the lowest amputation rate of 2.63% is achieved with the pathway revascularization surgery → lipid lowering → antiplatelet. This shows that surgical interventions, when combined with pharmacological therapies, provide superior outcomes for HLD patients, closely matching the lower end of general PAD outcomes (e.g., 1.3%).

Non-amputation outcomes are notably strong in HLD patients. The highest rate of 35.80% is achieved with antiplatelet followed by lipid lowering, outperforming the general PAD rate of 32% for the same sequence. However, reversing the sequence to lipid lowering followed by antiplatelet achieves a lower rate of 26.72%, which is still above the general PAD equivalent of 24.1%. These findings suggest that HLD patients benefit greatly from pharmacological management, with slightly improved outcomes compared to the general PAD population.

In the *All of Us* dataset, in the cohort of PAD with hyperlipidemia, we identified 4072 patients, out of which 170 patients experienced amputations, and 3902 patients experienced no amputations. Figure 14 is a Sankey diagram of treatment sequences in the PAD with hyperlipidemia cohort in our *All of Us* dataset. Appendix A illustrate the dominant pathways that lead to amputation and non-amputation, which can be observed from the figure.

In the *All of Us* dataset, HLD patients experience higher amputation rates when treatments are less comprehensive. For instance, endovascular revascularization alone results in a rate of 14.12%, which is slightly higher than the general PAD rate of 13.6% for the same pathway. However, integrating revascularization surgery reduces the rate significantly to 5.88%, and the most effective pathway, antiplatelet → lipid lowering → endovascular revascularization, achieves a rate of 2.94%. This is slightly worse than the general PAD rate of 2.84% for a similar sequence but demonstrates the effectiveness of comprehensive treatment in HLD patients.

Non-amputation outcomes for HLD patients in the *All of Us* dataset are competitive with general PAD data. The highest rate of 29% is observed for lipid lowering followed by antiplatelet, which closely matches the general PAD outcome of 27.29% for the same sequence. Less effective pathways, such as antiplatelet alone and lipid lowering alone, yield non-amputation rates of 20% and 15%, respectively, which are slightly worse than the general PAD rates of 23% and 14%.

#### 3.2.7. Analysis of Treatment Pathways in PAD Patients with Age > 50

In the STARR dataset, in the cohort of patients older than 50 years, we identified 5313 patients, out of which 74 patients experienced amputations, and 5239 patients experienced no amputations. Figure 15 is a Sankey diagram of treatment sequences in the PAD filtered for patients who are aged over 50 cohort in our STARR dataset. Appendix A illustrate the dominant pathways that lead to amputation and non-amputation, which can be observed from the figure.

In the STARR dataset, age over 50 significantly impacts PAD treatment outcomes, particularly in amputation pathways. For instance, lipid lowering alone results in the highest amputation rate of 20.25%, which is substantially worse than the general PAD rate of 7.79% for the worst-performing sequence. However, introducing combination therapies leads to better outcomes. For example, antiplatelet → lipid lowering → endovascular revascularization reduces the amputation rate to 7.59%, which is comparable to general PAD outcomes for multi-modal treatments. The most effective sequence, revascularization surgery → lipid lowering → antiplatelet, achieves the lowest amputation rate of l2.53%. This highlights the importance of integrating surgical interventions and comprehensive pharmacological management to mitigate the increased risks associated with age.

For non-amputation outcomes, the highest rate of 32.43% is achieved with antiplatelet followed by lipid lowering, closely aligning with the general PAD rate of 32% for the same sequence. However, reversing the sequence to lipid lowering followed by antiplatelet results in a non-amputation rate of 24.31%, which is slightly better than the general PAD rate of 24.1%. This suggests that while combination therapies remain effective for older patients, age-related complications might necessitate prioritizing certain sequences for optimal outcomes.

In the *All of Us* dataset, in the cohort of patients older than 50 years, we identified 3416 patients, out of which 140 patients experienced amputations, and 3276 patients experienced no amputations. Figure 16 is a Sankey diagram of treatment sequences in the PAD with age > 50 cohort in our *All of Us* dataset. Appendix A illustrate the dominant pathways that lead to amputation and non-amputation, which can be observed from the figure.

In the *All of Us* dataset, endovascular revascularization alone leads to a high amputation rate of 17.14%, higher than the general PAD rate of 13.6%, indicating that age over 50 increases risks with standalone surgical treatments. However, combining therapies significantly improves outcomes. For instance, lipid lowering → antiplatelet → endovascular revascularization reduces the rate to 4.29%, and antiplatelet → lipid lowering → endovascular revascularization further lowers it to 3.57%. The best results are achieved with revascularization surgery alone, yielding an amputation rate of 2.86%.

For non-amputation outcomes, the highest rate of 28.41% is observed with lipid lowering → antiplatelet, slightly exceeding the general PAD rate of 27.29%. Other sequences, such as antiplatelet → lipid lowering, achieve 21.05%, which is comparable to the general PAD rate of 22%. These findings suggest that age over 50 does not drastically alter non-amputation outcomes when combination therapies are applied but does highlight the need for sequencing optimization to address age-related challenges.

#### 3.2.8. Analysis of Treatment Pathways in PAD Patients with Age ≤ 50

In the STARR dataset, in the cohort of patients aged 50 years or younger, we identified 268 patients, out of which 3 patients experienced amputations, and 265 patients experienced no amputations. Figure 17 is a Sankey diagram of treatment sequences in the PAD filtered for patients with age ≤ 50 cohort in our STARR dataset. Appendix A illustrate the dominant pathways that lead to amputation and non-amputation, which can be observed from the figure.

For patients aged 50 and younger in the STARR dataset, PAD treatment pathways show markedly higher amputation rates compared to the general PAD population. Antiplatelet followed by lipid lowering results in an exceptionally high amputation rate of 66.67%. This highlights a poor response to pharmacological therapies alone in younger patients, potentially indicating more aggressive disease progression or a lack of timely intervention. A simpler sequence, antiplatelet alone, fares slightly better with an amputation rate of 33.33%.

Non-amputation outcomes in this age group are also less favorable compared to the general PAD population. The highest non-amputation rate, 18.49%, is achieved with lipid lowering alone, which is higher than the general PAD rate of 16.8% for this sequence. Adding antiplatelet to lipid lowering slightly reduces the non-amputation rate to 17.36% and introducing endovascular revascularization after antiplatelet yields only 5.28%.

In the *All of Us* dataset, in the cohort of patients aged 50 years or younger, we identified 845 patients, out of which 36 patients experienced amputations, and 809 patients experienced no amputations. Figure 18 is a Sankey diagram of treatment sequences in the PAD with age ≤ 50 cohort in our *All of Us* dataset. Appendix A illustrate the dominant pathways that lead to amputation and non-amputation, which can be observed from the figure.

In the *All of Us* dataset, younger patients show similarly elevated amputation rates, though outcomes are slightly better than in the STARR dataset. Lipid lowering followed by antiplatelet achieves an amputation rate of 27.78%, which, while high, is significantly better than the 66.67% observed in the STARR dataset. The reverse sequence, antiplatelet followed by lipid lowering, further reduces the rate to 25%. Non-amputation outcomes are somewhat better in the *All of Us* dataset than in STARR. Antiplatelet alone achieves the highest non-amputation rate of 29%, comparable to the general PAD rate of 23%. Meanwhile, lipid lowering alone achieves 21.51%, which is better than the 18.49% seen in STARR for the same sequence and also higher than the general PAD rate of 14%. These results suggest that younger patients in the *All of Us* dataset respond better to standalone therapies than their counterparts in the STARR dataset.

#### 3.2.9. Analysis of Treatment Pathways in PAD Patients with Age > 65

In the STARR dataset, in the cohort of patients older than 65 years, we identified 3700 patients, out of which 51 patients experienced amputations, and 3649 patients experienced no amputations. Figure 19 is a Sankey diagram of treatment sequences in the PAD filtered for patients with age > 65 cohort in our STARR dataset. Appendix A illustrate the dominant pathways that lead to amputation and non-amputation, which can be observed from the figure.

In the STARR dataset, older PAD patients aged over 65 show significantly higher amputation rates for certain treatment pathways compared to the general PAD population. For example, lipid lowering followed by antiplatelet results in an amputation rate of 31.37%. Integrating surgical interventions proves more effective, as seen with antiplatelet → lipid lowering → endovascular revascularization, which reduces the amputation rate to 5.88%. While this is better than the higher rates seen in the general PAD rate of 7.79% for similar combinations, it suggests that age-related factors may increase risks even with optimal treatment.

Non-amputation outcomes in older patients show better alignment with general PAD data. The highest rate of 31.64% is achieved with antiplatelet followed by lipid lowering, which is comparable to the general PAD rate of 32% for this sequence. However, standalone treatments, such as lipid lowering alone, achieve only 18.21%, which is slightly better than the general PAD rate of 16.8% for the same approach. These results suggest that while combination therapies remain effective for older patients, age-related factors may reduce the efficacy of standalone strategies.

In the *All of Us* dataset, in the cohort of patients older than 65 years, we identified 1206 patients, out of which 54 patients experienced amputations, and 1152 patients experienced no amputations. Figure 20 is a Sankey diagram of treatment sequences in the PAD with age > 65 cohort in our *All of Us* dataset. Appendix A illustrate the dominant pathways that lead to amputation and non-amputation, which can be observed from the figure.

In the *All of Us* dataset, older patients face elevated amputation risks, particularly with standalone treatments. Endovascular revascularization alone leads to an amputation rate of 24.07%, significantly higher than the general PAD rate of 13.6%, highlighting the added risks of surgical intervention in this age group without supporting therapies. However, the combination of endovascular revascularization → revascularization surgery reduces the rate to 9.26%, and the best outcome is achieved with revascularization surgery alone, yielding an amputation rate of 5.56%. While this rate is close to the general PAD minimum of 2.84%, it still underscores the heightened risks associated with age.

Non-amputation outcomes in the *All of Us* dataset show that combination therapies are crucial for older patients. Lipid lowering followed by antiplatelet achieves the highest non-amputation rate of 29.89%, which is slightly better than the general PAD rate of 27.29% for the same sequence. However, simpler strategies, such as lipid lowering alone, yield a non-amputation rate of 18.27%, which is only marginally better than the general PAD rate of 14%. These results emphasize that while older patients can achieve comparable outcomes with well-designed treatment pathways, standalone treatments are less effective and riskier.

#### 3.2.10. Analysis of Treatment Pathways in PAD Patients with Age ≤ 65

In the STARR dataset, in the cohort of patients aged 65 years or younger, we identified 1881 patients, out of which 26 patients experienced amputations, and 1855 patients experienced no amputations. Figure 21 is a Sankey diagram of treatment sequences in the PAD filtered for patients with age ≤ 65 cohort in our STARR dataset. Appendix A illustrate the dominant pathways that lead to amputation and non-amputation, which can be observed from the figure.

In the STARR dataset, PAD patients aged 65 and younger show elevated amputation rates for standalone treatments compared to the general PAD population. For instance, lipid lowering alone results in the highest amputation rate of 24.14%. Multi-modal treatments prove far more effective. Both antiplatelet → lipid lowering → endovascular revascularization and antiplatelet → revascularization surgery → lipid lowering achieve an amputation rate of 6.9%, which is only slightly higher than the general PAD rate of 5.19% for similar sequences. These findings suggest that while younger patients may face aggressive disease progression, combination therapies help to mitigate these risks.

Non-amputation outcomes for younger patients in the STARR dataset are promising when combination therapies are used. The highest non-amputation rate, 31.86%, is achieved with antiplatelet followed by lipid lowering, closely matching the general PAD rate of 32% for the same sequence. However, less comprehensive strategies, such as lipid lowering followed by antiplatelet, result in a non-amputation rate of 18.87%, which is slightly lower than the general PAD rate of 24.1%. This indicates that while younger patients can achieve favorable outcomes with pharmacological strategies, sequencing and the inclusion of surgical interventions play a crucial role in optimizing results.

In the *All of Us* dataset, in the cohort of patients aged 65 years or younger, we identified 3055 patients, out of which 122 patients experienced amputations, and 2933 patients experienced no amputations. Figure 22 is a Sankey diagram of treatment sequences in the PAD with age ≤ 65 cohort in our *All of Us* dataset. Appendix A illustrate the dominant pathways that lead to amputation and non-amputation, which can be observed from the figure.

In the *All of Us* dataset, endovascular revascularization alone results in an amputation rate of 10.92%, which is better than the 24.14% observed in STARR for lipid lowering alone and also the general PAD rate of 13.6%. The combination of endovascular revascularization followed by revascularization surgery significantly reduces the rate to 5.04%, while the best-performing sequence, lipid lowering → antiplatelet → endovascular revascularization, achieves an amputation rate of 4.2%, closely aligning with the general PAD rate of 3.98%.

Non-amputation outcomes in the *All of Us* dataset highlight the effectiveness of combination therapies. Lipid lowering followed by antiplatelet produces the highest non-amputation rate of 24.92%, which is slightly lower than the general PAD rate of 27.29% for this sequence. Meanwhile, antiplatelet alone achieves a non-amputation rate of 24.48%, comparable to the general PAD rate of 23%. However, less comprehensive strategies, such as lipid lowering alone, yield a rate of 18.51%, which is higher than the general PAD rate of 14%. These results suggest that younger patients in the *All of Us* dataset respond better to treatments than their counterparts in STARR.

#### 3.2.11. Analysis of Treatment Pathways in PAD Patients with Age ≤ 80

In the STARR dataset, in the cohort of patients aged 80 years or younger, we identified 4613 patients, out of which 58 patients experienced amputations, and 4555 patients experienced no amputations. Figure 23 is a Sankey diagram of treatment sequences in the PAD filtered for patients with age ≤ 80 cohort in our STARR dataset. Appendix A illustrate the dominant pathways that lead to amputation and non-amputation, which can be observed from the figure.

In the STARR dataset, PAD patients aged 80 and younger show elevated amputation rates for standalone treatments compared to the general PAD population. For instance, lipid lowering alone results in an amputation rate of 18.03%. Multi-modal treatments prove more effective. Both antiplatelet → revascularization surgery → lipid lowering and antiplatelet → lipid lowering → endovascular revascularization achieve an amputation rate of 6.56%, closely aligning with the general PAD rate of 5.19% for similar sequences. These findings suggest that while older patients may face aggressive disease progression, combination therapies may help to mitigate these risks effectively.

Non-amputation outcomes for patients aged 80 and younger in the STARR dataset remain promising when combination therapies are used. The highest non-amputation rate, 31.94%, is achieved with antiplatelet → lipid lowering, closely matching the general PAD rate of 32% for the same sequence. However, less comprehensive strategies, such as lipid lowering → antiplatelet, result in a non-amputation rate of 22.74%, which is slightly lower than the general PAD rate of 24.1%. This indicates that while older patients can achieve favorable outcomes with pharmacological strategies, sequencing and the inclusion of surgical interventions remain critical to optimizing results.

In the *All of Us* dataset, in the cohort of patients aged 80 years or younger, we identified 4223 patients, out of which 174 patients experienced amputations, and 4049 patients experienced no amputations. Figure 24 is a Sankey diagram of treatment sequences in the PAD filtered for patients with age ≤ 80 cohort in our *All of Us* dataset. Appendix A illustrate the dominant pathways that lead to amputation and non-amputation, which can be observed from the figure.

In the *All of Us* dataset, endovascular revascularization alone results in an amputation rate of 13.71%, which is slightly better than the 18.03% observed in STARR for lipid lowering alone but still worse than the general PAD rate of 13.6%. The combination of endovascular revascularization → revascularization surgery significantly reduces the rate to 5.71%, while the best-performing sequence, lipid lowering → antiplatelet → endovascular revascularization, achieves an amputation rate of 4.0%, closely aligning with the general PAD rate of 3.98%. These findings emphasize the importance of multi-modal therapies in reducing amputation risk for older patients.

Non-amputation outcomes in the *All of Us* dataset highlight the effectiveness of combination therapies. Lipid lowering → antiplatelet produces the highest non-amputation rate of 27.15%, closely matching the general PAD rate of 27.29% for this sequence. Meanwhile, standalone antiplatelet achieves a non-amputation rate of 23.18%, slightly better than the general PAD rate of 23%. Less comprehensive strategies, such as lipid lowering alone, yield a non-amputation rate of 14.78%, which is slightly higher than the general PAD rate of 14%. These results suggest that older patients in the *All of Us* dataset respond favorably to treatments and, in some cases, outperform their counterparts in the STARR dataset.

#### 3.2.12. Analysis of Treatment Pathways in PAD Patients with Age > 80

In the STARR dataset, in the cohort of patients aged over 80 years, we identified 1583 patients, out of which 19 patients experienced amputations, and 1564 patients experienced no amputations. Figure 25 is a Sankey diagram of treatment sequences in the PAD filtered for patients with age > 80 cohort in our STARR dataset. Appendix A illustrate the dominant pathways that lead to amputation and non-amputation, which can be observed from the figure.

In the STARR dataset, PAD patients aged over 80 years exhibit significantly elevated amputation rates for specific treatment sequences compared to the general PAD population. For instance, lipid lowering → antiplatelet results in the highest amputation rate of 42.11%. A less common sequence, lipid lowering → antiplatelet → endovascular revascularization, achieves an amputation rate of 10.53%. These findings suggest that older patients may experience more aggressive disease progression and require advanced treatment strategies to mitigate these risks.

Non-amputation outcomes for patients aged over 80 in the STARR dataset are encouraging when combination therapies are used. The highest non-amputation rate, 30.98%, is achieved with antiplatelet → lipid lowering, closely aligning with the general PAD rate of 32%. Standalone lipid lowering yields a non-amputation rate of 22.75%, slightly lower than the 24.1% observed in the general PAD population. These results indicate that older patients benefit from structured pharmacological strategies, but standalone approaches may have limited effectiveness.

In the *All of Us* dataset, in the cohort of patients aged over 80 years, we identified 218 patients, out of which 2 patients experienced amputations, and 216 patients experienced no amputations. Figure 26 is a Sankey diagram of treatment sequences in PAD filtered for patients with age > 80 cohort in our *All of Us* dataset. Appendix A illustrate the dominant pathways that lead to amputation and non-amputation, which can be observed from the figure.

In the *All of Us* dataset, amputation outcomes for PAD patients aged over 80 are limited but severe. Both endovascular revascularization → revascularization surgery and revascularization surgery alone result in amputation rates of 50.0%. While the sample size is small, these results suggest that surgical interventions, when used alone, may not adequately prevent amputations in the oldest patients.

Non-amputation outcomes in the *All of Us* dataset highlight the value of combination therapies for patients over 80. The highest non-amputation rate, 30.52%, is achieved with lipid lowering → antiplatelet, closely matching the general PAD rate of 27.29%. Standalone lipid lowering yields a non-amputation rate of 23.0%, significantly higher than the general PAD rate of 14%. These findings suggest that older patients in the *All of Us* dataset respond well to pharmacological strategies, particularly when treatments are sequenced effectively.

#### 3.2.13. Analysis of Treatment Pathways in PAD Patients Who Reported Their Race as ‘White’

In the STARR dataset, in the cohort of patients who reported their race as white, we identified 3343 patients, out of which 39 patients experienced amputations, and 3304 patients experienced no amputations. Figure 27 is a Sankey diagram of treatment sequences in the PAD filtered for patients who reported their race as ‘white’ cohort in our STARR dataset. Appendix A illustrate the dominant pathways that lead to amputation and non-amputation, which can be observed from the figure.

In the STARR dataset, white patients with PAD exhibit amputation rates that are similar to the general PAD population when comprehensive treatments are applied. For example, antiplatelet → lipid lowering → endovascular revascularization and antiplatelet → revascularization surgery → lipid lowering both result in an amputation rate of 7.69%, closely aligning with the general PAD rate of 7.79% for similar sequences. The most effective treatment pathway, revascularization surgery → lipid lowering → antiplatelet, achieves the lowest amputation rate of 5.13%, which is higher than the general PAD rate of 2.6%. These findings suggest that multi-modal strategies are equally effective for white patients as they are for the general PAD population, with surgical interventions playing a crucial role in reducing amputation risks.

For non-amputation pathways, the highest rate of 31.84% is observed for antiplatelet followed by lipid lowering, comparable to the general PAD rate of 32% for the same sequence. However, less comprehensive approaches, such as lipid lowering followed by antiplatelet, yield a lower non-amputation rate of 23.15%, which is comparable to the general PAD rate of 24.1%. Standalone treatments like lipid lowering alone produce a non-amputation rate of 15.38%, comparable to the general PAD rate of 16.8%. These results indicate that white patients respond well to combination therapies, achieving outcomes on par with or slightly better than the general PAD population.

In the *All of Us* dataset, in the cohort of patients who reported their race as white, we identified 2515 patients, out of which 78 patients experienced amputations, and 2437 patients experienced no amputations. Figure 28 is a Sankey diagram of treatment sequences in the PAD filtered for patients who reported their race as ‘white’ cohort in our *All of Us* dataset. Appendix A illustrate the dominant pathways that lead to amputation and non-amputation, which can be observed from the figure.

In the *All of Us* dataset, amputation rates for white patients are slightly worse than the general PAD population for standalone treatments but improve significantly with combination therapies. For instance, endovascular revascularization alone results in an amputation rate of 15.38%, higher than the general PAD rate of 13.6%. Adding revascularization surgery reduces the rate to 8.97%, and the most effective sequence, lipid lowering → antiplatelet → endovascular revascularization, achieves an amputation rate of 3.85%, closely aligning with the general PAD rate of 3.98%.

Non-amputation outcomes in the *All of Us* dataset is competitive with general PAD results. The highest rate of 29.59% is achieved with lipid lowering followed by antiplatelet, slightly better than the general PAD rate of 27.29% for the same sequence. Other combinations, such as antiplatelet alone or antiplatelet followed by lipid lowering, yield non-amputation rates of 22.61% and 20.72%, respectively, which are slightly lower than the general PAD rates of 23% and 22%. Standalone lipid lowering achieves a rate of 15.80%, comparable to the general PAD rate of 14%, suggesting that white patients respond well to pharmacological interventions.

#### 3.2.14. Analysis of Treatment Pathways in PAD Patients Who Reported Their Race as ‘Black’

In the STARR dataset, in the cohort of patients who reported their race as black, we identified 311 patients, out of which 14 patients experienced amputations, and 297 patients experienced no amputations. Figure 29 is a Sankey diagram of treatment sequences in the PAD patients who reported their race as ‘black’ cohort in our STARR dataset. Appendix A illustrate the dominant pathways that lead to amputation and non-amputation, which can be observed from the figure.

In the STARR dataset, black patients with PAD experience significantly higher amputation rates compared to the general PAD population, particularly with standalone treatments. For instance, lipid lowering alone results in an amputation rate of 21.43%. Combination therapies, such as antiplatelet → lipid lowering → endovascular revascularization, reduce the amputation rate to 14.29%, which is still notably higher than the general PAD rate of 7.79% for similar sequences. These findings highlight the elevated risks faced by black patients with PAD and the need for more aggressive treatment strategies.

Non-amputation rates for black patients show relatively strong outcomes when combination therapies are used. For example, lipid lowering → antiplatelet achieves the highest non-amputation rate of 31.65%, which is higher than the general PAD rate of 24.1%. However, reversing the sequence to antiplatelet → lipid lowering produces a slightly lower non-amputation rate of 29.29%, which is still competitive with the general PAD rate of 32%. Simpler strategies, such as antiplatelet alone, yield a non-amputation rate of 10.77%, competitive to the general PAD rate of 10.2%.

In the *All of Us* dataset, in the cohort of patients who reported their race as black, we identified 1076 patients, out of which 67 patients experienced amputations, and 1009 patients experienced no amputations. Figure 30 is a Sankey diagram of treatment sequences in the PAD patients who reported their race as ‘black’ cohort in our *All of Us* dataset. Appendix A illustrate the dominant pathways that lead to amputation and non-amputation, which can be observed from the figure.

In the *All of Us* dataset, amputation rates for black patients are slightly better than in STARR but still worse than the general PAD population. Endovascular revascularization alone produces an amputation rate of 7.46%, which is lower than the general PAD rate of 13.6%. More comprehensive combinations, such as lipid lowering → antiplatelet → endovascular revascularization, achieve an amputation rate of 4.48%, aligning closely with the general PAD rate of 3.98%. A similar outcome is achieved with the addition of revascularization surgery to the sequence, maintaining an amputation rate of 4.48%.

Non-amputation outcomes in the *All of Us* dataset are competitive with the general PAD population. Antiplatelet alone achieves the highest non-amputation rate of 27.65%, slightly outperforming the general PAD rate of 23%. Other combinations, such as antiplatelet → lipid lowering, yield a non-amputation rate of 24.18%, which is slightly higher than the general PAD rate of 22%. Standalone lipid lowering produces a non-amputation rate of 11.69%, lower than the general PAD rate of 14%. These results suggest that while black patients benefit from combination therapies, they face greater challenges with standalone or less comprehensive treatments.

#### 3.2.15. Analysis of Treatment Pathways in PAD Patients Who Reported Their Race as ‘Asian’

In the STARR dataset, in the cohort of patients who reported their race as Asian, we identified 708 patients, out of which 4 patients experienced amputations, and 704 patients experienced no amputations. Figure 31 is a Sankey diagram of treatment sequences in the PAD patients who reported their race as ‘Asian’ cohort in our STARR dataset. Appendix A illustrate the dominant pathways that lead to amputation and non-amputation, which can be observed from the figure.

In the STARR dataset, counts of amputations in Asian patients with PAD are extremely low, but from whatever we observed in the samples, it can be noted that in both treatment sequences—lipid lowering → antiplatelet and antiplatelet → lipid lowering—result in amputation rates of 50%. Non-amputation outcomes in the STARR dataset are also notably poor for Asian patients. Lipid lowering alone achieves the highest non-amputation rate of 16.76%, which is lower than the general PAD rate of 16.8% for the same treatment. Antiplatelet alone produces a dismal non-amputation rate of 7.67%, lower than the general PAD rate of 10.2%. These findings indicate that not only are standalone treatments ineffective in preventing amputations in Asian patients, but even non-amputation outcomes fall well short of general PAD benchmarks.

In the *All of Us* dataset, in the cohort of patients who reported their race as Asian, we identified 40 patients, out of which 1 patient experienced an amputation, and 39 patients experienced no amputations. Figure 32 is a Sankey diagram of treatment sequences in PAD patients who reported their race as ‘Asian’ cohort in our *All of Us* dataset. Appendix A illustrate the dominant pathways that lead to amputation and non-amputation, which can be observed from the figure.

The *All of Us* dataset paints an even grimmer picture for Asian patients, with just one patient in the cohort and endovascular revascularization alone resulting in the amputation. The highest non-amputation rate of 33.33% is achieved with both lipid lowering → antiplatelet and antiplatelet → lipid lowering, exceeding the general PAD rate of 27.29% for similar sequences. Standalone lipid lowering produces a non-amputation rate of 23.08%, which is also slightly better than the general PAD rate of 14%. These results may suggest that while surgical interventions fail to deliver positive outcomes, pharmacological strategies remain effective in improving non-amputation outcomes for Asian patients, but the extremely low sample size is important to be considered.

#### 3.2.16. Analysis of Treatment Pathways in PAD Patients Who Reported Their Gender as ‘Male’

In the STARR dataset, in the male patient cohort, we identified 2251 patients, out of which 56 patients experienced amputations, and 2195 patients experienced no amputations. Figure 33 is a Sankey diagram of treatment sequences in PAD patients who reported their gender as ‘male’ cohort in our STARR dataset. Appendix A illustrate the dominant pathways that lead to amputation and non-amputation, which can be observed from the figure.

In the STARR dataset, male PAD patients show slightly elevated amputation rates compared to the general PAD population for similar treatment pathways. For instance, antiplatelet → lipid lowering → endovascular revascularization results in an amputation rate of 8.93%, which is marginally higher than the general PAD rate of 7.79% for the same sequence. Similarly, antiplatelet → revascularization surgery → lipid lowering achieves an amputation rate of 7.14%, slightly better than the general PAD rate of 5.19%. These findings suggest that while combination therapies remain effective for male patients, their outcomes are slightly worse than those seen in the general population, likely due to increased disease severity or other risk factors.

Non-amputation outcomes for male patients in the STARR dataset are slightly better than in the general PAD population for some treatment sequences. For example, antiplatelet → lipid lowering achieves the highest non-amputation rate of 33.45%, comparable to the general PAD rate of 32%. However, simpler sequences, such as lipid lowering → antiplatelet, yield a lower non-amputation rate of 25.32%, which is still comparable to the general PAD rate of 24.1%. Standalone lipid lowering achieves 16.40%, aligning closely with the general PAD rate of 16.8%. These results indicate that male patients respond well to combination therapies, with outcomes closely mirroring those of the general population.

In the *All of Us* dataset, in the male patient cohort, we identified 1994 patients, out of which 105 patients experienced amputations, and 1889 patients experienced no amputations. Figure 34 is a Sankey diagram of treatment sequences in the PAD patients who reported their gender as ‘male’ cohort in our *All of Us* dataset. Appendix A illustrate the dominant pathways that lead to amputation and non-amputation, which can be observed from the figure.

In the *All of Us* dataset, amputation rates for male patients are slightly elevated for standalone treatments compared to the general PAD population. For instance, endovascular revascularization alone produces an amputation rate of 14.29%, comparable to the general PAD rate of 13.6%. Adding revascularization surgery reduces the rate to 7.62%, which aligns closely with the general PAD rate of 6.25%. These results highlight the importance of combination therapies in reducing amputation risks for male patients, though outcomes remain slightly worse than in the general PAD population.

Non-amputation outcomes in the *All of Us* dataset for male patients are competitive with the general PAD population. The highest rate of 29.89% is achieved with lipid lowering → antiplatelet, which is slightly higher than the general PAD rate of 27.29%. Other sequences, such as antiplatelet → lipid lowering and antiplatelet alone, yield non-amputation rates of 21.33% and 21.44%, closely matching the general PAD rates of 22% and 23%, respectively. These findings suggest that while standalone treatments are less effective, pharmacological strategies remain robust for male patients.

#### 3.2.17. Analysis of Treatment Pathways in PAD Patients Who Reported Their Gender as ‘Female’ Cohort

In the STARR dataset, in the female patient cohort, we identified 2137 patients, out of which 21 patients experienced amputations, and 2116 patients experienced no amputations. Figure 35 is a Sankey diagram of treatment sequences in the PAD patients who reported their gender as ‘female’ cohort in our STARR dataset. Appendix A illustrate the dominant pathways that lead to amputation and non-amputation, which can be observed from the figure.

In the STARR dataset, female PAD patients experience significantly higher amputation rates for standalone or less comprehensive treatments compared to the general PAD population. For instance, lipid lowering → antiplatelet results in an alarmingly high amputation rate of 47.62%. However, combination therapies, such as antiplatelet → exercise therapy → lipid lowering and endovascular revascularization → lipid lowering, achieve substantially lower amputation rates of 4.76%. This underscores the critical importance of multi-modal approaches, particularly for female patients, to mitigate amputation risks effectively.

Non-amputation outcomes for female patients in the STARR dataset show encouraging results with combination therapies. For example, antiplatelet → lipid lowering achieves a non-amputation rate of 30.86%, which is slightly below the general PAD rate of 32% for the same sequence. Simpler pathways, such as lipid lowering alone or antiplatelet alone, result in lower non-amputation rates of 17.77% and 12.71%, respectively, compared to the general PAD rates of 16.8% and 10.2%. These results suggest that while combination therapies remain effective for female patients, standalone treatments are considerably less effective in preventing disease progression.

In the *All of Us* dataset, in the female patient cohort, we identified 2129 patients, out of which 66 patients experienced amputations, and 2063 patients experienced no amputations. Figure 36 is a Sankey diagram of treatment sequences in the PAD patients who reported their gender as ‘female’ cohort in our *All of Us* dataset. Appendix A illustrate the dominant pathways that lead to amputation and non-amputation, which can be observed from the figure.

In the *All of Us* dataset, female patients also show elevated amputation rates for standalone treatments compared to the general PAD population. For example, lipid lowering → antiplatelet results in an amputation rate of 27.27%. However, endovascular revascularization alone achieves a reduced amputation rate of 13.68%, which is close to the general PAD rate of 13.6%, suggesting that surgical interventions may have similar efficacy for female patients as for the general population.

Non-amputation outcomes in the *All of Us* dataset for female patients show slightly better results than the general PAD population. Antiplatelet alone achieves a non-amputation rate of 25.06%, which is slightly better than the general PAD rate of 23%. Combination therapies, such as antiplatelet → lipid lowering, achieve 22.78%, higher than the general PAD rate of 22%. Standalone lipid lowering produces a non-amputation rate of 16.48%, slightly better than the general PAD rate of 14%. These findings highlight the importance of combination therapies for female patients while suggesting that even standalone pharmacological strategies can provide moderate benefits.

#### 3.2.18. Analysis of Treatment Pathways in PAD Patients Who Reportedly Smoke

In the STARR dataset, in the cohort of patients who smoke, we identified 5215 patients, out of which 72 patients experienced amputations, and 5143 patients experienced no amputations. Figure 37 is a Sankey diagram of treatment sequences in the PAD patients who reportedly smoke cohort in our STARR dataset. Appendix A illustrate the dominant pathways that lead to amputation and non-amputation, which can be observed from the figure.

In the STARR dataset, PAD patients who smoke experience amputation rates that closely align with the general PAD population when combination therapies are applied. For instance, antiplatelet → lipid lowering → endovascular revascularization results in an amputation rate of 8.33%, slightly higher than the general PAD rate of 7.79% for the same sequence. Similarly, antiplatelet → revascularization surgery → lipid lowering achieves an amputation rate of 5.56%, comparable to the general PAD rate of 5.19%. These results indicate that multi-modal approaches remain effective in managing PAD in smokers, though their outcomes are marginally worse compared to non-smokers, likely due to the compounding effects of smoking on vascular health.

Non-amputation outcomes for smokers in the STARR dataset are also comparable to the general PAD population. Antiplatelet → lipid lowering achieves the highest non-amputation rate of 32.14%, close to the general PAD rate of 32% for the same sequence. However, simpler pathways, such as lipid lowering → antiplatelet, and lipid lowering alone, yield non-amputation rates of 24.62% and 16.45%, which align closely with the general PAD rates of 24.1% and 16.8%, respectively. These findings suggest that smokers benefit from combination therapies, achieving outcomes similar to the general PAD population when multi-modal strategies are used.

In the *All of Us* dataset, in the cohort of patients who smoke, we identified 1283 patients, out of which 78 patients experienced amputations, and 1205 patients experienced no amputations. Figure 38 is a Sankey diagram of treatment sequences in the PAD patients who reportedly smoke cohort in our *All of Us* dataset. Appendix A illustrate the dominant pathways that lead to amputation and non-amputation, which can be observed from the figure.

In the *All of Us* dataset, amputation rates for smokers are slightly worse than the general PAD population for standalone treatments but improve significantly with combination therapies. Endovascular revascularization alone produces an amputation rate of 11.54%, which is lower than the general PAD rate of 13.6%. Adding revascularization surgery reduces the rate to 6.41%, and the best outcome is achieved with antiplatelet → lipid lowering → endovascular revascularization, which results in an amputation rate of 3.85%, closely aligning with the general PAD rate of 3.98%.

Non-amputation outcomes in the *All of Us* dataset for smokers show slightly lower results compared to the general PAD population. The highest non-amputation rate of 24.90% is achieved with lipid lowering → antiplatelet, which is below the general PAD rate of 27.29% for the same sequence. Other pathways, such as antiplatelet alone or antiplatelet → lipid lowering, yield non-amputation rates of 22.78% and 21%, respectively, which are lower than the general PAD rates of 23% and 22%. These results indicate that while combination therapies remain effective, smoking may reduce the overall efficacy of certain treatment pathways.

### 3.3. Summary from Confounder Analysis

In the STARR dataset, out of 5581 PAD patients, 77 underwent amputation. Among the risk factors assessed, hyperlipidemia was the most prevalent in the amputation group, with 76 patients affected. This represents a substantial proportion compared to other comorbidities, such as diabetes (54 amputations) and hypertension (56 amputations). Notably, heart failure and cerebrovascular disease were associated with fewer amputations, with 26 cases each. The comparison among the risk factors between STARR and *All of Us* can be observed in Figure 39.

The high incidence of amputations in the hyperlipidemia subset underscores its critical role in the progression of PAD leading to limb loss. Similarly, in the *All of Us* dataset comprising 4261 PAD patients, 176 experienced amputations. Hyperlipidemia was again one of the leading risk factors, with 170 amputations reported. Other significant comorbidities include hypertension (173 amputations) and diabetes (160 amputations). Heart failure and cerebrovascular disease were associated with fewer amputations, with 106 and 117 cases, respectively, as seen in Figure 35. The consistency of hyperlipidemia as a leading risk factor across both cohorts highlights its pervasive impact on PAD outcomes. Certain PAD confounder groups, such as patients identifying as Asian, exhibit negligible amputation counts, likely due to their low representation in both the datasets. The comparison of patient counts with and without amputations among various racial groups can be found in Figure 40.

Endovascular revascularization treatment is notably present in the heart failure cohort of the STARR dataset, accounting for 20% of the total amputation pathways, while its occurrence has not been found in the dominant non-amputation pathways. This treatment also appears prominently in the cerebrovascular disease and coronary artery disease cohorts, with at least 15% occurrence in both risk factors within the *All of Us* dataset, while being absent in the dominant non-amputation pathways. Revascularization procedures have also been found to be more prominent in the full PAD cohorts of both STARR and *All of Us* data, confirming the findings from the confounder analysis. Antiplatelet and lipid-lowering treatments are prominent across all the cohorts, often routing to non-amputation just like the treatments of smoking cessation and exercise therapy as seen in the Sankey visualizations. The prevalence of these treatment sequences varies by slight percentages across different risk factors.

Amputation outcomes in PAD patients across the STARR and *All of Us* datasets reveal significant variations based on confounders, underscoring the need for tailored interventions. Hypertension confounders exhibit promising results with multi-modal approaches, such as revascularization surgery → lipid lowering → antiplatelet → endovascular revascularization, achieving an amputation rate of 1.79% in the STARR dataset. Similarly, diabetic patients show favorable outcomes with sequences like lipid lowering → antiplatelet → endovascular revascularization, yielding a low 3.7% amputation rate. On the other hand, patients with heart failure face higher risks, with antiplatelet → lipid lowering → endovascular revascularization resulting in a significantly elevated rate of 19.23% in STARR. Racial disparities are evident, with black patients experiencing elevated risks; for instance, lipid lowering alone results in a 21.43% amputation rate (STARR). Similarly, Asian patients appear to demonstrated amputation risk for standalone endovascular revascularization in the *All of Us* dataset, but it is to be noted that their sample size is negligible as shown in Table 12 and Table 13. Smoking, while slightly elevating risks, does not drastically affect outcomes when combination therapies like antiplatelet → lipid lowering → endovascular revascularization are employed, achieving rates closely aligned with the general PAD population. Age-related confounders also show stark differences, with older patients (>65 years) responding better to aggressive therapies, such as endovascular revascularization → revascularization surgery (9.26% in *All of Us*), while younger patients (<50 years) face disproportionately high risks, such as a 66.67% rate for antiplatelet → lipid lowering in STARR, highlighting the need for more aggressive and early interventions in younger cohorts. However, the younger patient count is very low in STARR data as represented in Figure 41; on the basis of patient counts in cohorts with an age ≤ 50, age > 50, age < 65, age ≥ 65, age < 80, age ≥ 80, we derived the patient counts for age 51–65, age 66–80.

Non-amputation outcomes across these risk factors emphasize the effectiveness of pharmacological and combination therapies in mitigating disease progression, particularly in patients with hypertension, diabetes, and smoking history. For hypertensive patients, antiplatelet → lipid lowering yields a robust 33% non-amputation rate in STARR, slightly exceeding the general PAD rate of 32%. Similarly, diabetic patients benefit significantly from this sequence, achieving 31% in STARR. Smokers and hyperlipidemia patients also show strong results with lipid lowering → antiplatelet combinations, achieving non-amputation rates of 31.65% (STARR) and 29.59% (*All of Us*), respectively. However, standalone treatments are markedly less effective, with rates often dropping below 15% for lipid lowering alone across most confounders, particularly in heart failure (7.43% in STARR) and older patients (>65 years; 18.21% in STARR). Racial disparities persist in the *All of Us* dataset as well, though pharmacological strategies demonstrate some promise for black and Asian patients, with sequences like lipid lowering → antiplatelet achieving rates of 27.65% and 33.33%, respectively. These findings underscore the importance of prioritizing combination therapies tailored to individual confounders to improve outcomes, as even minor variations in sequencing can drastically affect both amputation and non-amputation rates, highlighting the need for precision medicine in PAD management.

## 4. Conclusions and Future Work

The dominant pathways for amputations in both the datasets used include sequences often with revascularization procedures, antiplatelet therapy, and lipid-lowering medication. Additionally, pathways involving surgical revascularization, either alone or in combination with lipid-lowering medications, showed a high association with amputation outcomes. Building on the findings of this study, future work could explore the development of predictive models using advanced machine learning techniques. One promising avenue is the application of natural language processing (NLP) models to treatment sequences. By leveraging the power of NLP, we can create models that predict the next likely treatment or outcome based on historical sequences.

Such an NLP model would analyze the treatment pathways as sequences of events, learning the patterns and associations within the data. This approach could help in the prediction of patient outcomes and provide valuable decision support for clinicians. Additionally, future research may focus on integrating more comprehensive datasets, including patient demographics, comorbidities, and lifestyle factors, to enhance the predictive accuracy of these models. By considering a broader range of variables, we can develop more robust models that better reflect the complexity of real-world clinical scenarios, that could predict the likelihood of various future outcomes, allowing for more proactive and personalized treatment planning. Further statistical analysis and validation of the identified significant pathways across diverse patient populations and settings would also be beneficial. This could help to generalize the findings and ensure their applicability in different clinical contexts. Moreover, investigating the cost-effectiveness and long-term outcomes of these treatment pathways could provide valuable insights into optimizing healthcare resources and improving patient care.

In conclusion, while this study has provided insights into treatment pathways and their associations with amputation and non-amputation outcomes, there is ample opportunity for future work to build on these findings. By developing predictive models and conducting further research, we can continue to advance our understanding of cardiovascular treatment strategies and enhance patient outcomes.

## Figures and Tables

**Figure 1 biomedicines-13-00258-f001:**
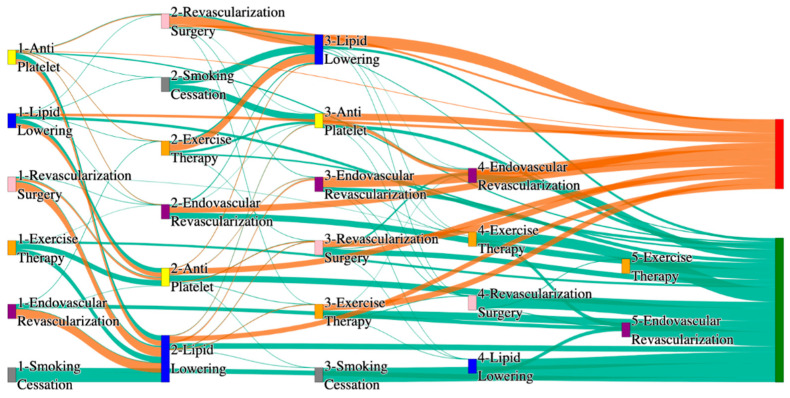
Sankey plot for the STARR data, depicting the flow of treatments between the patients that experienced amputation and those who did not experience any amputation. Thickness of the pathway indicates the percentage of patients experiencing the treatment. The size of the amputation cohort is visually larger here due to the data normalized at the cohort and node levels; cohort normalization adjusts sequence counts by total cohort size, while node normalization ensures flow values sum to 1 at each node, enabling visual comparison of pathway proportions as stated in the Section 2.3. Pathways colored in deep amber are the amputation pathways and those colored in pine green are non-amputation pathways.

**Figure 2 biomedicines-13-00258-f002:**
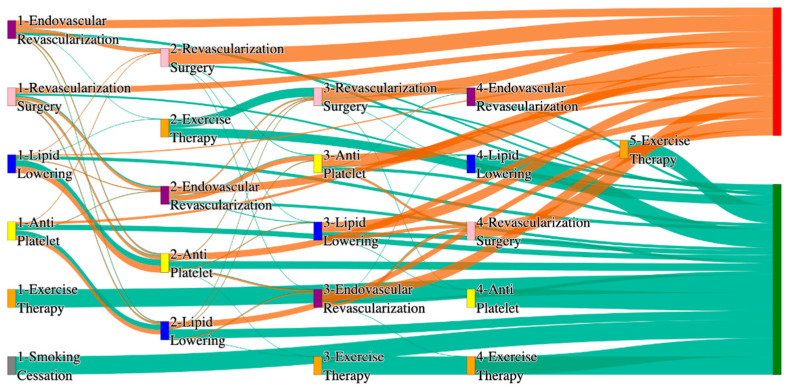
Sankey plot for the *All of Us* data, depicting the flow of treatments between the patients that experienced amputation and those who did not experience any amputation. Thickness of the pathway indicates the percentage of patients experiencing the treatment. The size of the amputation cohort is visually larger here due to the data normalized at the cohort and node levels; cohort normalization adjusts sequence counts by total cohort size, while node normalization ensures flow values sum to 1 at each node, enabling visual comparison of pathway proportions as stated in the Section 2.3. Pathways colored in deep amber are the amputation pathways and those colored in pine green are non-amputation pathways.

**Figure 3 biomedicines-13-00258-f003:**
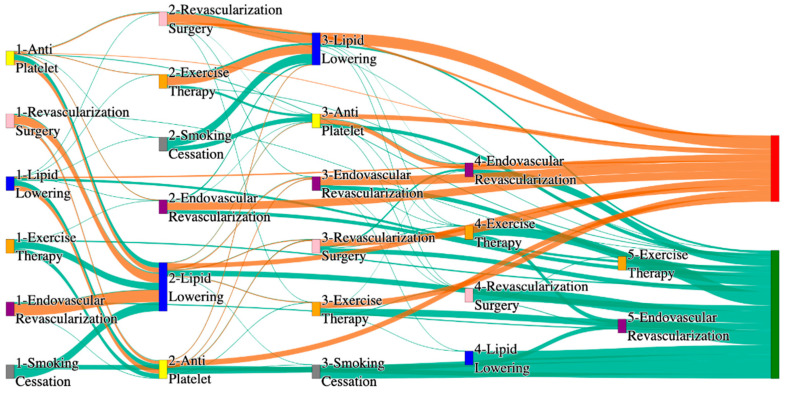
Sankey plot for the PAD with hypertension in STARR data, depicting the flow of treatments between the patients that experienced amputation and those who did not experience any amputation. Thickness of the pathway indicates the percentage of patients experiencing the treatment. The size of the amputation cohort is visually larger here due to the data normalized at the cohort and node levels; cohort normalization adjusts sequence counts by total cohort size, while node normalization ensures flow values sum to 1 at each node, enabling visual comparison of pathway proportions as stated in the Section 2.3. Pathways colored in deep amber are the amputation pathways and those colored in pine green are non-amputation pathways.

**Figure 4 biomedicines-13-00258-f004:**
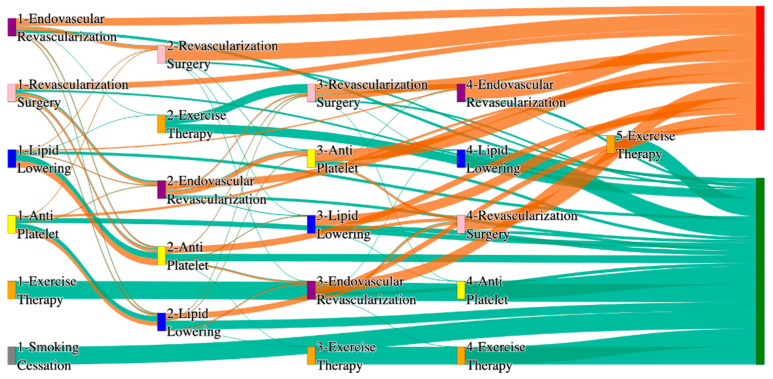
Sankey plot for the PAD with hypertension in *All of Us* data, depicting the flow of treatments between the patients that experienced amputation and those who did not experience any amputation. Thickness of the pathway indicates the percentage of patients experiencing the treatment. The size of the amputation cohort is visually larger here due to the data normalized at the cohort and node levels; cohort normalization adjusts sequence counts by total cohort size, while node normalization ensures flow values sum to 1 at each node, enabling visual comparison of pathway proportions as stated in the Section 2.3. Pathways colored in deep amber are the amputation pathways and those colored in pine green are non-amputation pathways.

**Figure 5 biomedicines-13-00258-f005:**
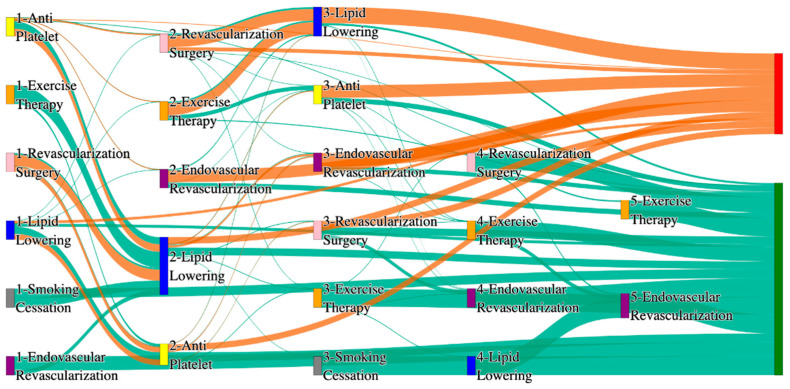
Sankey plot for the PAD with diabetes in STARR data, depicting the flow of treatments between the patients that experienced amputation and those who did not experience any amputation. Thickness of the pathway indicates the percentage of patients experiencing the treatment. The size of the amputation cohort is visually larger here due to the data normalized at the cohort and node levels; cohort normalization adjusts sequence counts by total cohort size, while node normalization ensures flow values sum to 1 at each node, enabling visual comparison of pathway proportions as stated in the Section 2.3. Pathways colored in deep amber are the amputation pathways and those colored in pine green are non-amputation pathways.

**Figure 6 biomedicines-13-00258-f006:**
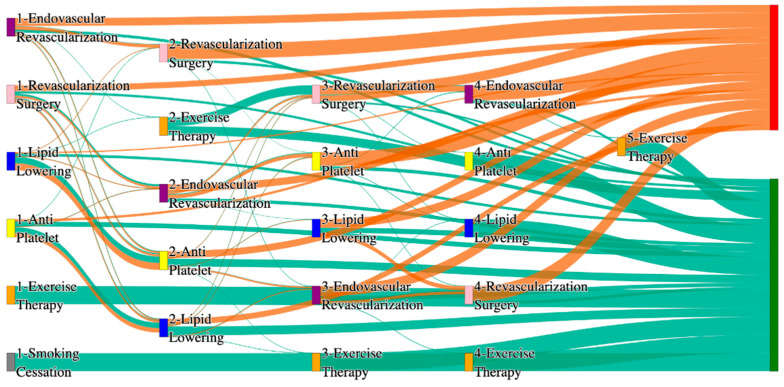
Sankey plot for the PAD with diabetes in *All of Us* data, depicting the flow of treatments between the patients that experienced amputation and those who did not experience any amputation. Thickness of the pathway indicates the percentage of patients experiencing the treatment. The size of the amputation cohort is visually larger here due to the data normalized at the cohort and node levels; cohort normalization adjusts sequence counts by total cohort size, while node normalization ensures flow values sum to 1 at each node, enabling visual comparison of pathway proportions as stated in the Section 2.3. Pathways colored in deep amber are the amputation pathways and those colored in pine green are non-amputation pathways.

**Figure 7 biomedicines-13-00258-f007:**
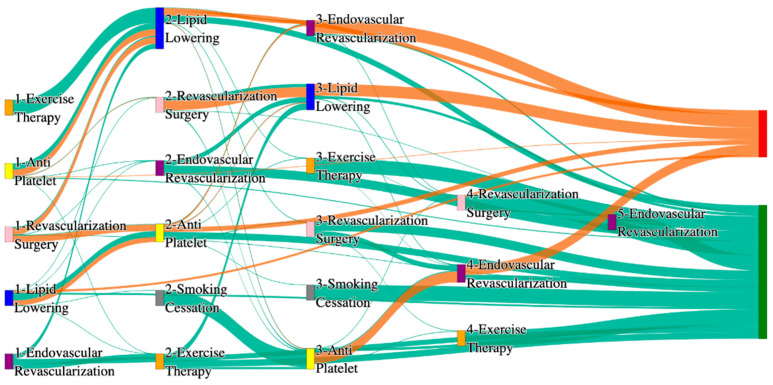
Sankey plot for the PAD with heart failure in STARR data, depicting the flow of treatments between the patients that experienced amputation and those who did not experience any amputation. Thickness of the pathway indicates the percentage of patients experiencing the treatment. The size of the amputation cohort is visually larger here due to the data normalized at the cohort and node levels; cohort normalization adjusts sequence counts by total cohort size, while node normalization ensures flow values sum to 1 at each node, enabling visual comparison of pathway proportions as stated in the Section 2.3. Pathways colored in deep amber are the amputation pathways and those colored in pine green are non-amputation pathways.

**Figure 8 biomedicines-13-00258-f008:**
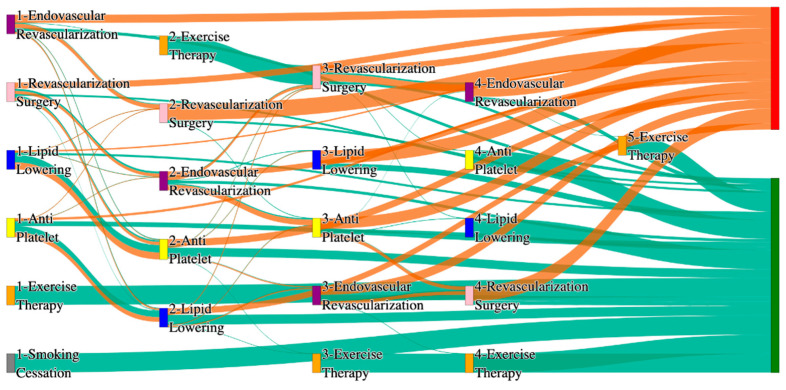
Sankey plot for the PAD with heart failure in *All of Us* data, depicting the flow of treatments between the patients that experienced amputation and those who did not experience any amputation. Thickness of the pathway indicates the percentage of patients experiencing the treatment. The size of the amputation cohort is visually larger here due to the data normalized at the cohort and node levels; cohort normalization adjusts sequence counts by total cohort size, while node normalization ensures flow values sum to 1 at each node, enabling visual comparison of pathway proportions as stated in the Section 2.3. Pathways colored in deep amber are the amputation pathways and those colored in pine green are non-amputation pathways.

**Figure 9 biomedicines-13-00258-f009:**
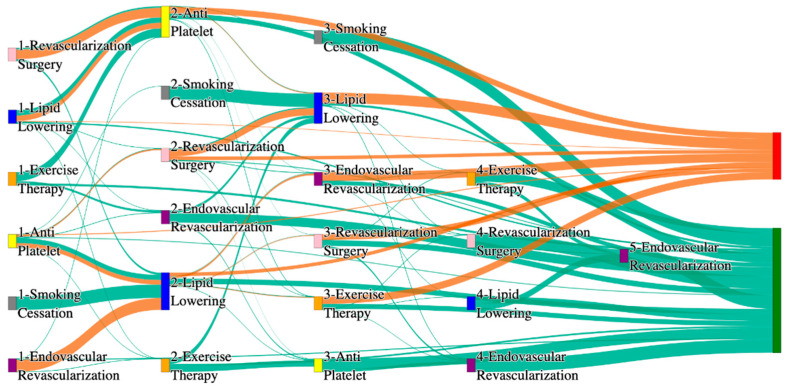
Sankey plot for the PAD with cerebrovascular disease in STARR data, depicting the flow of treatments between the patients that experienced amputation and those who did not experience any amputation. Thickness of the pathway indicates the percentage of patients experiencing the treatment. The size of the amputation cohort is visually larger here due to the data normalized at the cohort and node levels: cohort normalization adjusts sequence counts by total cohort size, while node normalization ensures flow values sum to 1 at each node, enabling visual comparison of pathway proportions as stated in the Section 2.3. Pathways colored in deep amber are the amputation pathways and those colored in pine green are non-amputation pathways.

**Figure 10 biomedicines-13-00258-f010:**
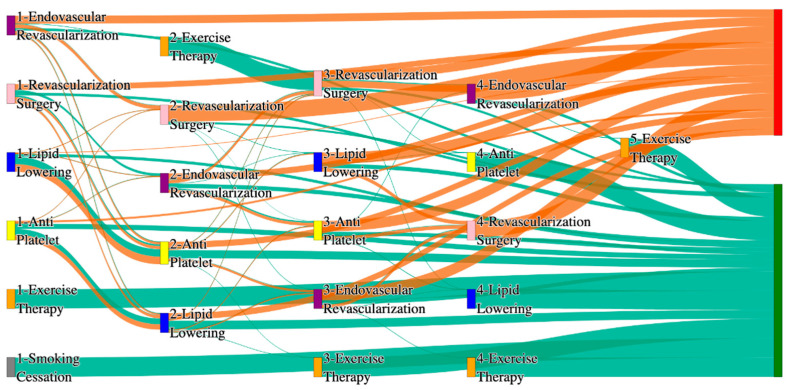
Sankey plot for the PAD with cerebrovascular disease in *All of Us* data, depicting the flow of treatments between the patients that experienced amputation and those who did not experience any amputation. Thickness of the pathway indicates the percentage of patients experiencing the treatment. The size of the amputation cohort is visually larger here due to the data normalized at the cohort and node levels; cohort normalization adjusts sequence counts by total cohort size, while node normalization ensures flow values sum to 1 at each node, enabling visual comparison of pathway proportions as stated in the Section 2.3. Pathways colored in deep amber are the amputation pathways and those colored in pine green are non-amputation pathways.

**Figure 11 biomedicines-13-00258-f011:**
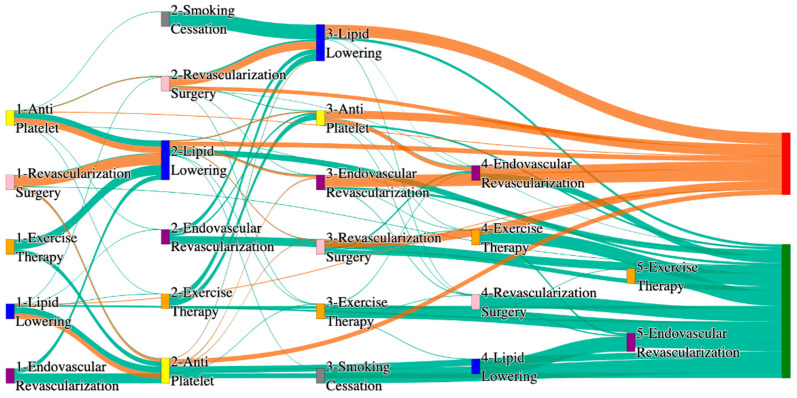
Sankey plot for the PAD with coronary artery disease in STARR data, depicting the flow of treatments between the patients that experienced amputation and those who did not experience any amputation. Thickness of the pathway indicates the percentage of patients experiencing the treatment. The size of the amputation cohort is visually larger here due to the data normalized at the cohort and node levels; cohort normalization adjusts sequence counts by total cohort size, while node normalization ensures flow values sum to 1 at each node, enabling visual comparison of pathway proportions as stated in the Section 2.3. Pathways colored in deep amber are the amputation pathways and those colored in pine green are non-amputation pathways.

**Figure 12 biomedicines-13-00258-f012:**
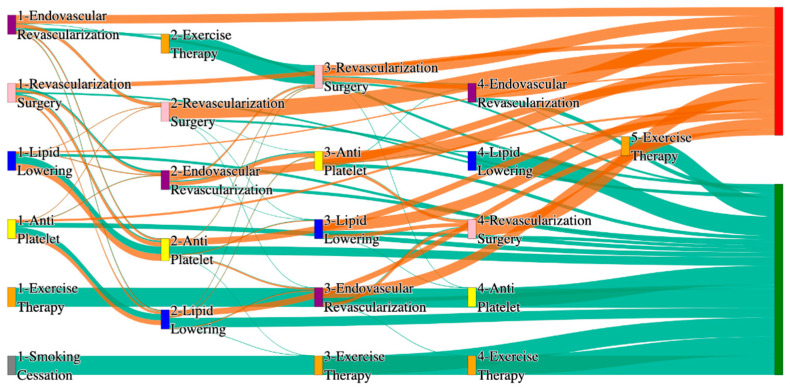
Sankey plot for the PAD with coronary artery disease in *All of Us* data, depicting the flow of treatments between the patients that experienced amputation and those who did not experience any amputation. Thickness of the pathway indicates the percentage of patients experiencing the treatment. The size of the amputation cohort is visually larger here due to the data normalized at the cohort and node levels; cohort normalization adjusts sequence counts by total cohort size, while node normalization ensures flow values sum to 1 at each node, enabling visual comparison of pathway proportions as stated in the Section 2.3. Pathways colored in deep amber are the amputation pathways and those colored in pine green are non-amputation pathways.

**Figure 13 biomedicines-13-00258-f013:**
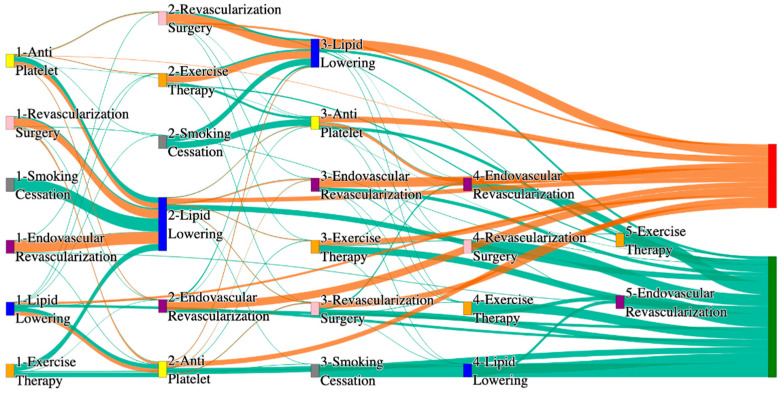
Sankey plot for the PAD with hyperlipidemia in STARR data, depicting the flow of treatments between the patients that experienced amputation and those who did not experience any amputation. Thickness of the pathway indicates the percentage of patients experiencing the treatment. The size of the amputation cohort is visually larger here due to the data normalized at the cohort and node levels; cohort normalization adjusts sequence counts by total cohort size, while node normalization ensures flow values sum to 1 at each node, enabling visual comparison of pathway proportions as stated in the Section 2.3. Pathways colored in deep amber are the amputation pathways and those colored in pine green are non-amputation pathways.

**Figure 14 biomedicines-13-00258-f014:**
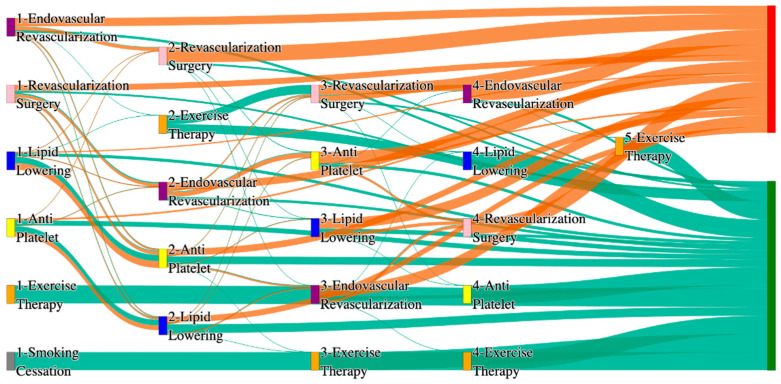
Sankey plot for the PAD with hyperlipidemia in *All of Us* data, depicting the flow of treatments between the patients that experienced amputation and those who did not experience any amputation. Thickness of the pathway indicates the percentage of patients experiencing the treatment. The size of the amputation cohort is visually larger here due to the data normalized at the cohort and node levels; cohort normalization adjusts sequence counts by total cohort size, while node normalization ensures flow values sum to 1 at each node, enabling visual comparison of pathway proportions as stated in the Section 2.3. Pathways colored in deep amber are the amputation pathways and those colored in pine green are non-amputation pathways.

**Figure 15 biomedicines-13-00258-f015:**
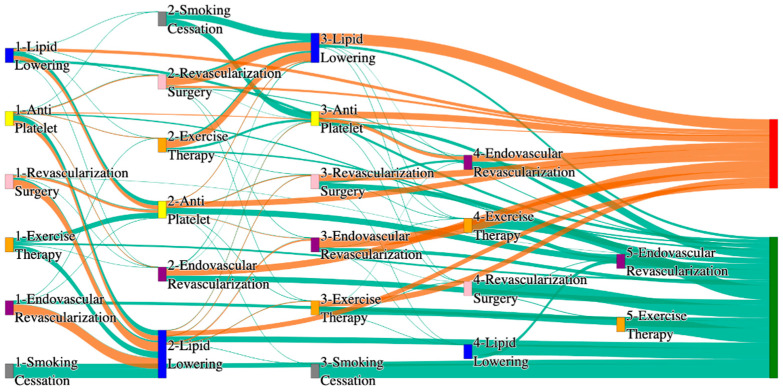
Sankey plot for the PAD with age > 50 in STARR data, depicting the flow of treatments between the patients that experienced amputation and those who did not experience any amputation. Thickness of the pathway indicates the percentage of patients experiencing the treatment. The size of the amputation cohort is visually larger here due to the data normalized at the cohort and node levels; cohort normalization adjusts sequence counts by total cohort size, while node normalization ensures flow values sum to 1 at each node, enabling visual comparison of pathway proportions as stated in the Section 2.3. Pathways colored in deep amber are the amputation pathways and those colored in pine green are non-amputation pathways.

**Figure 16 biomedicines-13-00258-f016:**
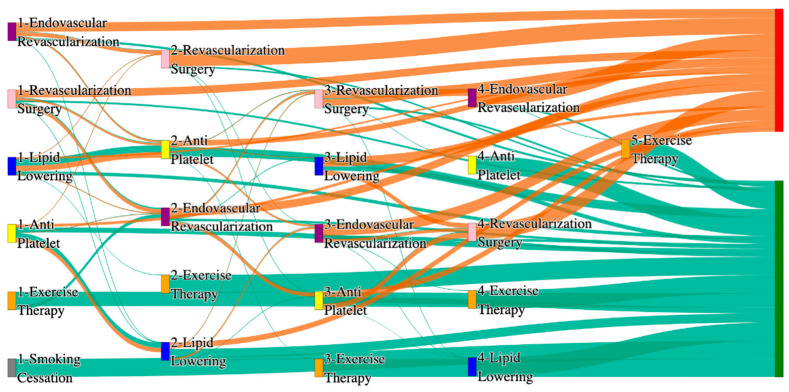
Sankey plot for the PAD with age > 50 in *All of Us* data, depicting the flow of treatments between the patients that experienced amputation and those who did not experience any amputation. Thickness of the pathway indicates the percentage of patients experiencing the treatment. The size of the amputation cohort is visually larger here due to the data normalized at the cohort and node levels; cohort normalization adjusts sequence counts by total cohort size, while node normalization ensures flow values sum to 1 at each node, enabling visual comparison of pathway proportions as stated in the Section 2.3. Pathways colored in deep amber are the amputation pathways and those colored in pine green are non-amputation pathways.

**Figure 17 biomedicines-13-00258-f017:**
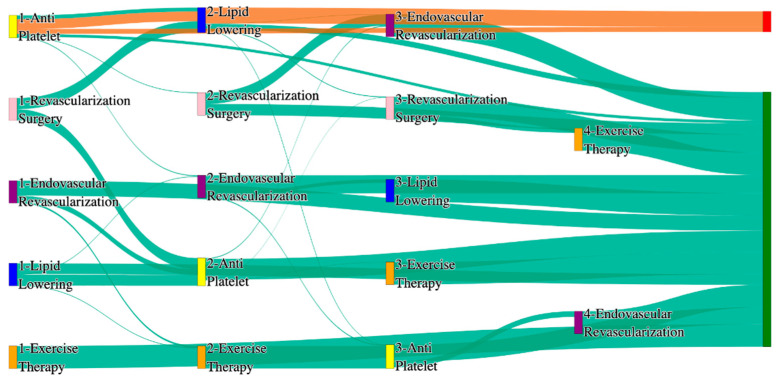
Sankey plot for the PAD with age ≤ 50 in STARR data, depicting the flow of treatments between the patients that experienced amputation and those who did not experience any amputation. Thickness of the pathway indicates the percentage of patients experiencing the treatment. The size of the amputation cohort is visually larger here due to the data normalized at the cohort and node levels; cohort normalization adjusts sequence counts by total cohort size, while node normalization ensures flow values sum to 1 at each node, enabling visual comparison of pathway proportions as stated in the Section 2.3. Pathways colored in deep amber are the amputation pathways and those colored in pine green are non-amputation pathways.

**Figure 18 biomedicines-13-00258-f018:**
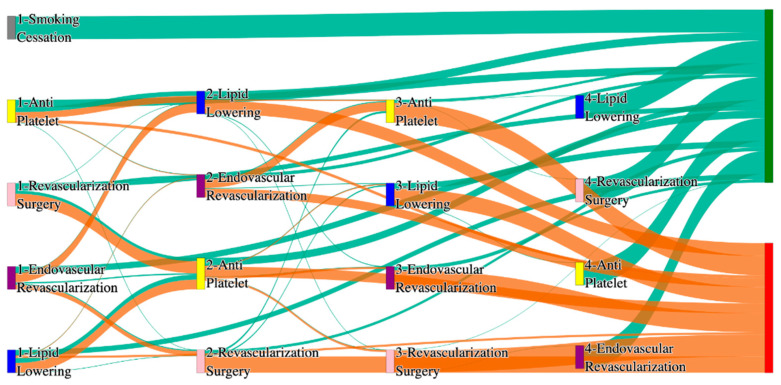
Sankey plot for the PAD with age ≤ 50 in *All of Us* data, depicting the flow of treatments between the patients that experienced amputation and those who did not experience any amputation. Thickness of the pathway indicates the percentage of patients experiencing the treatment. The size of the amputation cohort is visually larger here due to the data normalized at the cohort and node levels; cohort normalization adjusts sequence counts by total cohort size, while node normalization ensures flow values sum to 1 at each node, enabling visual comparison of pathway proportions as stated in the Section 2.3. Pathways colored in deep amber are the amputation pathways and those colored in pine green are non-amputation pathways.

**Figure 19 biomedicines-13-00258-f019:**
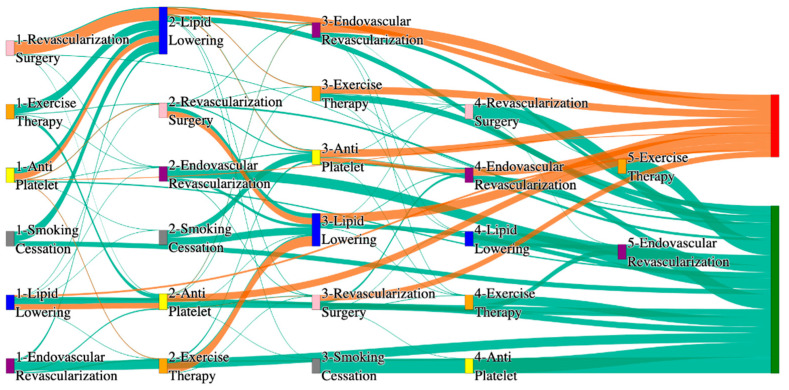
Sankey plot for the PAD with age > 65 in STARR data, depicting the flow of treatments between the patients that experienced amputation and those who did not experience any amputation. Thickness of the pathway indicates the percentage of patients experiencing the treatment. The size of the amputation cohort is visually larger here due to the data normalized at the cohort and node levels; cohort normalization adjusts sequence counts by total cohort size, while node normalization ensures flow values sum to 1 at each node, enabling visual comparison of pathway proportions as stated in the Section 2.3. Pathways colored in deep amber are the amputation pathways and those colored in pine green are non-amputation pathways.

**Figure 20 biomedicines-13-00258-f020:**
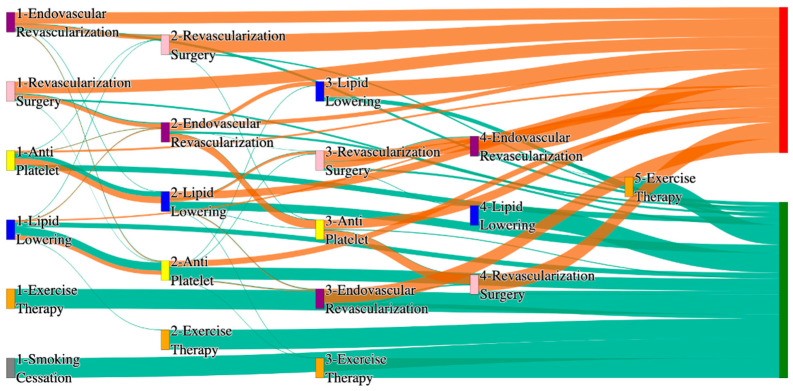
Sankey plot for the PAD with age > 65 in *All of Us* data, depicting the flow of treatments between the patients that experienced amputation and those who did not experience any amputation. Thickness of the pathway indicates the percentage of patients experiencing the treatment. The size of the amputation cohort is visually larger here due to the data normalized at the cohort and node levels; cohort normalization adjusts sequence counts by total cohort size, while node normalization ensures flow values sum to 1 at each node, enabling visual comparison of pathway proportions as stated in the Section 2.3. Pathways colored in deep amber are the amputation pathways and those colored in pine green are non-amputation pathways.

**Figure 21 biomedicines-13-00258-f021:**
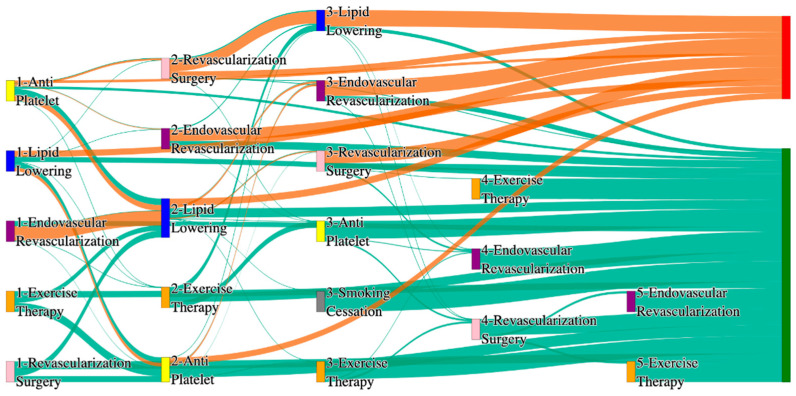
Sankey plot for the PAD with age ≤ 65 in STARR data, depicting the flow of treatments between the patients that experienced amputation and those who did not experience any amputation. Thickness of the pathway indicates the percentage of patients experiencing the treatment. The size of the amputation cohort is visually larger here due to the data normalized at the cohort and node levels; cohort normalization adjusts sequence counts by total cohort size, while node normalization ensures flow values sum to 1 at each node, enabling visual comparison of pathway proportions as stated in the Section 2.3. Pathways colored in deep amber are the amputation pathways and those colored in pine green are non-amputation pathways.

**Figure 22 biomedicines-13-00258-f022:**
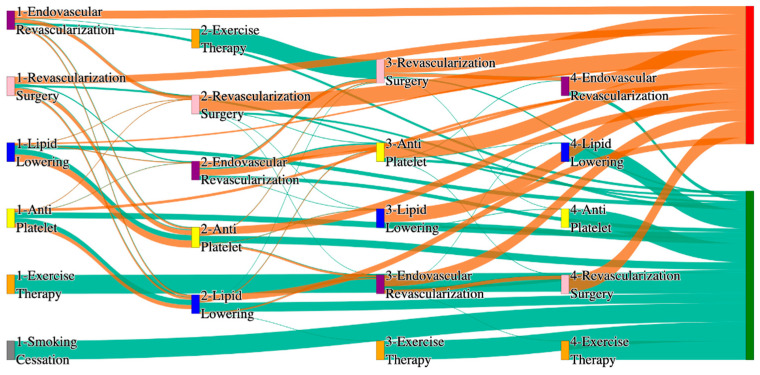
Sankey plot for the PAD with age ≤ 65 in *All of Us* data, depicting the flow of treatments between the patients that experienced amputation and those who did not experience any amputation. Thickness of the pathway indicates the percentage of patients experiencing the treatment. The size of the amputation cohort is visually larger here due to the data normalized at the cohort and node levels: cohort normalization adjusts sequence counts by total cohort size, while node normalization ensures flow values sum to 1 at each node, enabling visual comparison of pathway proportions as stated in the Section 2.3. Pathways colored in deep amber are the amputation pathways and those colored in pine green are non-amputation pathways.

**Figure 23 biomedicines-13-00258-f023:**
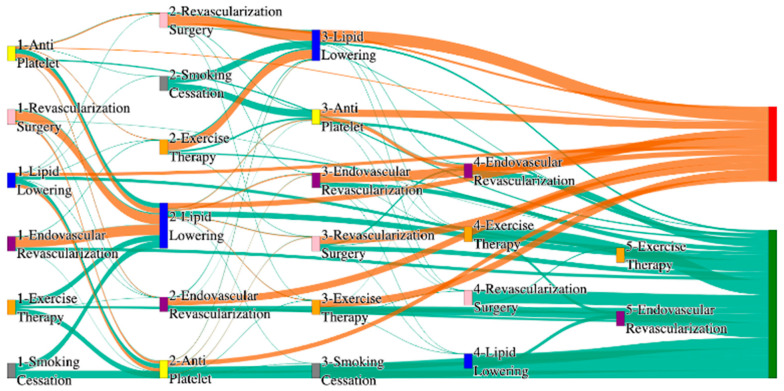
Sankey plot for the PAD with age ≤ 80 in STARR data, depicting the flow of treatments between the patients that experienced amputation and those who did not experience any amputation. Thickness of the pathway indicates the percentage of patients experiencing the treatment. The size of the amputation cohort is visually larger here due to the data normalized at the cohort and node levels; cohort normalization adjusts sequence counts by total cohort size, while node normalization ensures flow values sum to 1 at each node, enabling visual comparison of pathway proportions as stated in the Section 2.3. Pathways colored in deep amber are the amputation pathways and those colored in pine green are non-amputation pathways.

**Figure 24 biomedicines-13-00258-f024:**
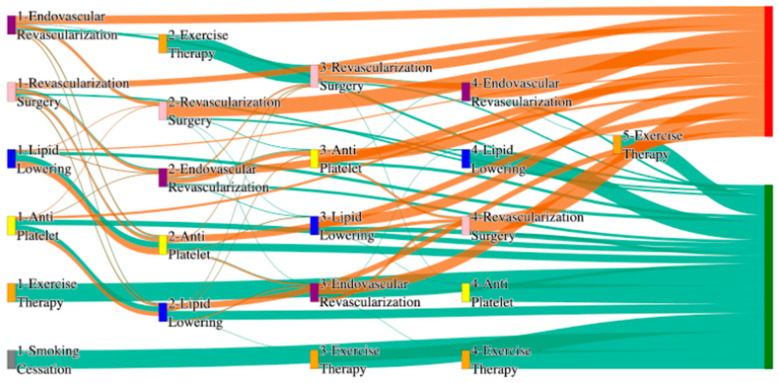
Sankey plot for the PAD with age ≤ 80 in *All of Us* data, depicting the flow of treatments between the patients that experienced amputation and those who did not experience any amputation. Thickness of the pathway indicates the percentage of patients experiencing the treatment. The size of the amputation cohort is visually larger here due to the data normalized at the cohort and node levels; cohort normalization adjusts sequence counts by total cohort size, while node normalization ensures flow values sum to 1 at each node, enabling visual comparison of pathway proportions as stated in the Section 2.3. Pathways colored in deep amber are the amputation pathways and those colored in pine green are non-amputation pathways.

**Figure 25 biomedicines-13-00258-f025:**
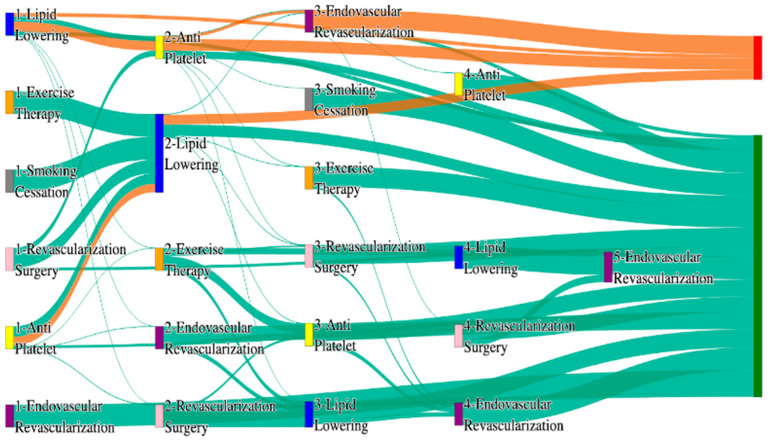
Sankey plot for the PAD with age > 80 in STARR data, depicting the flow of treatments between the patients that experienced amputation and those who did not experience any amputation. Thickness of the pathway indicates the percentage of patients experiencing the treatment. The size of the amputation cohort is visually larger here due to the data normalized at the cohort and node levels; cohort normalization adjusts sequence counts by total cohort size, while node normalization ensures flow values sum to 1 at each node, enabling visual comparison of pathway proportions as stated in the Section 2.3. Pathways colored in deep amber are the amputation pathways and those colored in pine green are non-amputation pathways.

**Figure 26 biomedicines-13-00258-f026:**
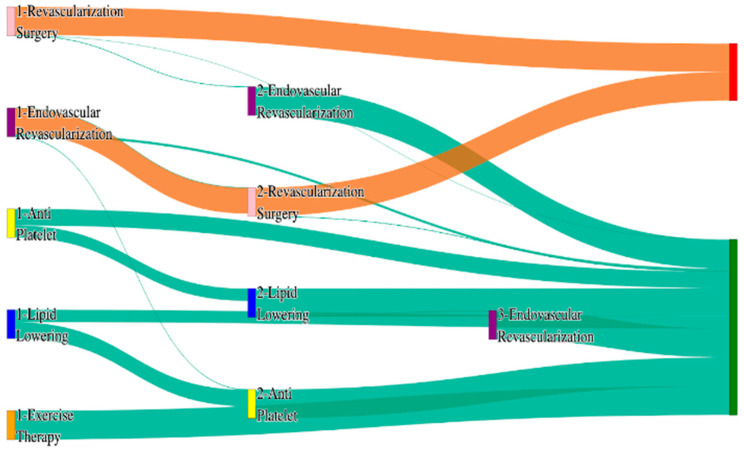
Sankey plot for the PAD with age > 80 in *All of Us* data, depicting the flow of treatments between the patients that experienced amputation and those who did not experience any amputation. Thickness of the pathway indicates the percentage of patients experiencing the treatment. The size of the amputation cohort is visually larger here due to the data normalized at the cohort and node levels; cohort normalization adjusts sequence counts by total cohort size, while node normalization ensures flow values sum to 1 at each node, enabling visual comparison of pathway proportions as stated in the Section 2.3. Pathways colored in deep amber are the amputation pathways and those colored in pine green are non-amputation pathways.

**Figure 27 biomedicines-13-00258-f027:**
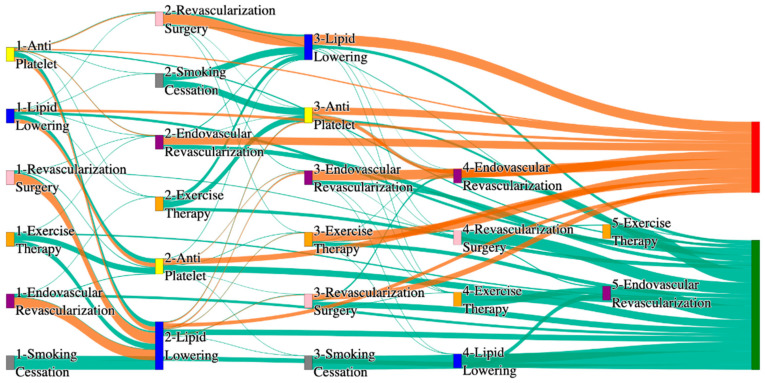
Sankey plot for the PAD filtered for patients who reported their race as ‘white’ cohort in STARR data, depicting the flow of treatments between the patients that experienced amputation and those who did not experience any amputation. Thickness of the pathway indicates the percentage of patients experiencing the treatment. The size of the amputation cohort is visually larger here due to the data normalized at the cohort and node levels; cohort normalization adjusts sequence counts by total cohort size, while node normalization ensures flow values sum to 1 at each node, enabling visual comparison of pathway proportions as stated in the Section 2.3. Pathways colored in deep amber are the amputation pathways and those colored in pine green are non-amputation pathways.

**Figure 28 biomedicines-13-00258-f028:**
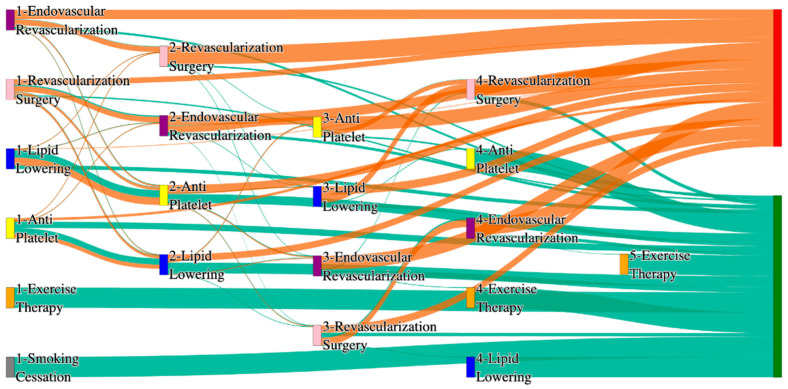
Sankey plot for the PAD filtered for patients who reported their race as ‘white’ cohort in *All of Us* data, depicting the flow of treatments between the patients that experienced amputation and those who did not experience any amputation. Thickness of the pathway indicates the percentage of patients experiencing the treatment. The size of the amputation cohort is visually larger here due to the data normalized at the cohort and node levels; cohort normalization adjusts sequence counts by total cohort size, while node normalization ensures flow values sum to 1 at each node, enabling visual comparison of pathway proportions as stated in the Section 2.3. Pathways colored in deep amber are the amputation pathways and those colored in pine green are non-amputation pathways.

**Figure 29 biomedicines-13-00258-f029:**
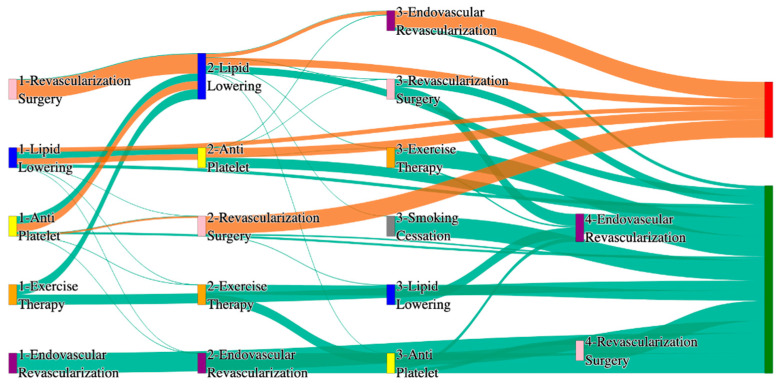
Sankey plot for the PAD patients who reported their race as ‘black’ cohort in STARR data, depicting the flow of treatments between the patients that experienced amputation and those who did not experience any amputation. Thickness of the pathway indicates the percentage of patients experiencing the treatment. The size of the amputation cohort is visually larger here due to the data normalized at the cohort and node levels; cohort normalization adjusts sequence counts by total cohort size, while node normalization ensures flow values sum to 1 at each node, enabling visual comparison of pathway proportions as stated in the Section 2.3. Pathways colored in deep amber are the amputation pathways and those colored in pine green are non-amputation pathways.

**Figure 30 biomedicines-13-00258-f030:**
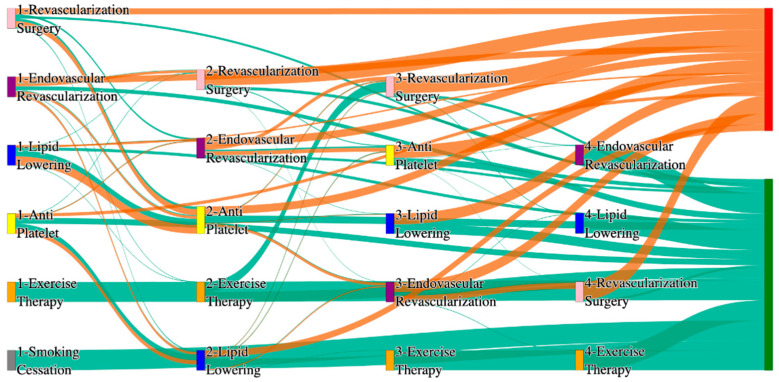
Sankey plot for the PAD patients who reported their race as ‘black’ cohort in *All of Us* data, depicting the flow of treatments between the patients that experienced amputation and those who did not experience any amputation. Thickness of the pathway indicates the percentage of patients experiencing the treatment. The size of the amputation cohort is visually larger here due to the data normalized at the cohort and node levels; cohort normalization adjusts sequence counts by total cohort size, while node normalization ensures flow values sum to 1 at each node, enabling visual comparison of pathway proportions as stated in the Section 2.3. Pathways colored in deep amber are the amputation pathways and those colored in pine green are non-amputation pathways.

**Figure 31 biomedicines-13-00258-f031:**
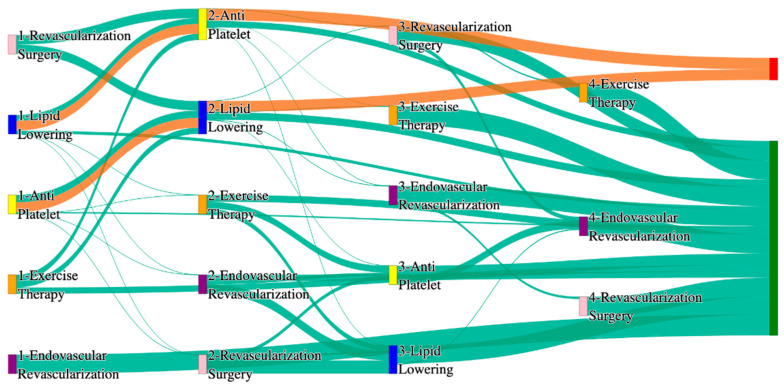
Sankey plot for the PAD patients who reported their race as ‘Asian’ cohort in STARR data, depicting the flow of treatments between the patients that experienced amputation and those who did not experience any amputation. Thickness of the pathway indicates the percentage of patients experiencing the treatment. The size of the amputation cohort is visually larger here due to the data normalized at the cohort and node levels; cohort normalization adjusts sequence counts by total cohort size, while node normalization ensures flow values sum to 1 at each node, enabling visual comparison of pathway proportions as stated in the Section 2.3. Pathways colored in deep amber are the amputation pathways and those colored in pine green are non-amputation pathways.

**Figure 32 biomedicines-13-00258-f032:**
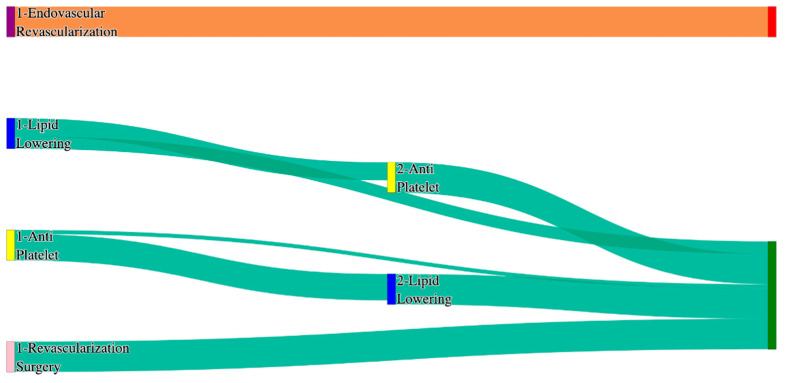
Sankey plot for the PAD patients who reported their race as ‘Asian’ cohort in *All of Us* data, depicting the flow of treatments between the patients that experienced amputation and those who did not experience any amputation. Thickness of the pathway indicates the percentage of patients experiencing the treatment. The size of the amputation cohort is visually larger here due to the data normalized at the cohort and node levels; cohort normalization adjusts sequence counts by total cohort size, while node normalization ensures flow values sum to 1 at each node, enabling visual comparison of pathway proportions as stated in the Section 2.3. Pathways colored in deep amber are the amputation pathways and those colored in pine green are non-amputation pathways.

**Figure 33 biomedicines-13-00258-f033:**
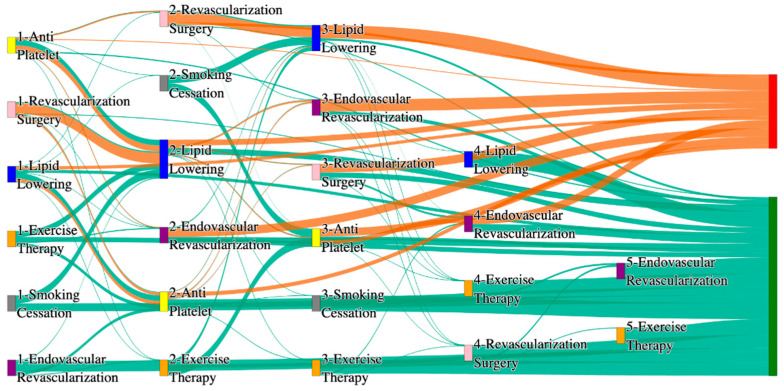
Sankey plot for the PAD patients who reported their gender as ‘male’ cohort in STARR data, depicting the flow of treatments between the patients that experienced amputation and those who did not experience any amputation. Thickness of the pathway indicates the percentage of patients experiencing the treatment. The size of the amputation cohort is visually larger here due to the data normalized at the cohort and node levels; cohort normalization adjusts sequence counts by total cohort size, while node normalization ensures flow values sum to 1 at each node, enabling visual comparison of pathway proportions as stated in the Section 2.3. Pathways colored in deep amber are the amputation pathways and those colored in pine green are non-amputation pathways.

**Figure 34 biomedicines-13-00258-f034:**
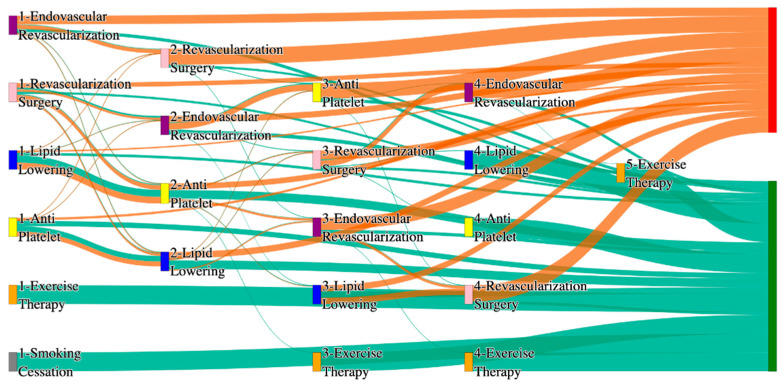
Sankey plot for the PAD patients who reported their gender as ‘male’ cohort in *All of Us* data, depicting the flow of treatments between the patients that experienced amputation and those who did not experience any amputation. Thickness of the pathway indicates the percentage of patients experiencing the treatment. The size of the amputation cohort is visually larger here due to the data normalized at the cohort and node levels; cohort normalization adjusts sequence counts by total cohort size, while node normalization ensures flow values sum to 1 at each node, enabling visual comparison of pathway proportions as stated in the Section 2.3. Pathways colored in deep amber are the amputation pathways and those colored in pine green are non-amputation pathways.

**Figure 35 biomedicines-13-00258-f035:**
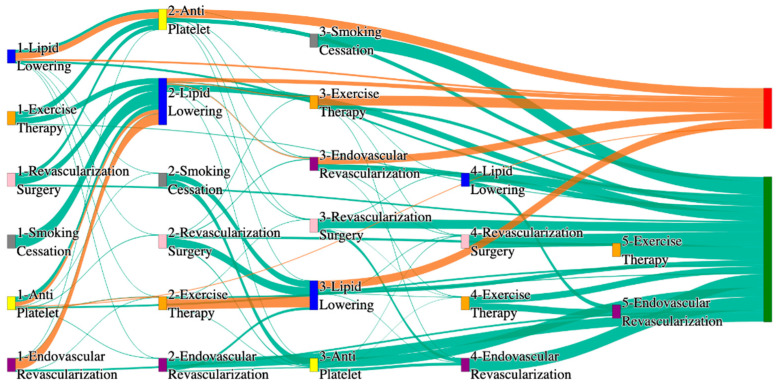
Sankey plot for the PAD patients who reported their gender as ‘Female’ cohort in STARR data, depicting the flow of treatments between the patients that experienced amputation and those who did not experience any amputation. Thickness of the pathway indicates the percentage of patients experiencing the treatment. The size of the amputation cohort is visually larger here due to the Data normalized at the cohort and node levels; cohort normalization adjusts sequence counts by total cohort size, while node normalization ensures flow values sum to 1 at each node, enabling visual comparison of pathway proportions as stated in the Section 2.3. Pathways colored in deep amber are the amputation pathways and those colored in pine green are non-amputation pathways.

**Figure 36 biomedicines-13-00258-f036:**
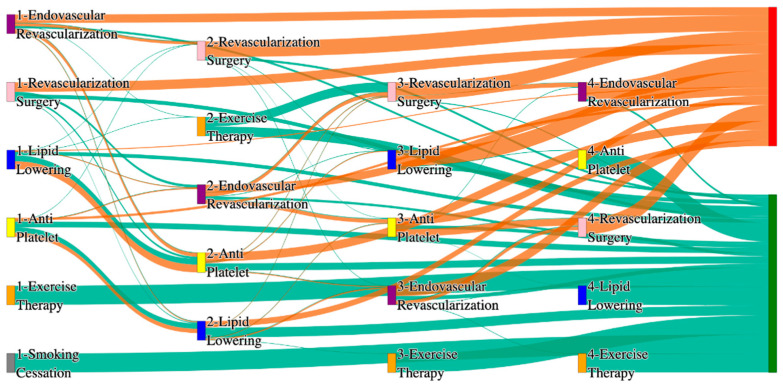
Sankey plot for the PAD patients who reported their gender as ‘female’ cohort in *All of Us* data, depicting the flow of treatments between the patients that experienced amputation and those who did not experience any amputation. Thickness of the pathway indicates the percentage of patients experiencing the treatment. The size of the amputation cohort is visually larger here due to the data normalized at the cohort and node levels; cohort normalization adjusts sequence counts by total cohort size, while node normalization ensures flow values sum to 1 at each node, enabling visual comparison of pathway proportions as stated in the Section 2.3. Pathways colored in deep amber are the amputation pathways and those colored in pine green are non-amputation pathways.

**Figure 37 biomedicines-13-00258-f037:**
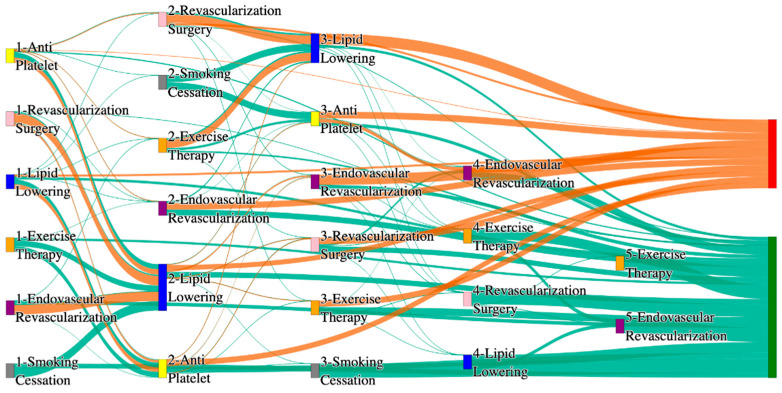
Sankey plot for the PAD patients who reportedly smoke in STARR data, depicting the flow of treatments between the patients that experienced amputation and those who did not experience any amputation. Thickness of the pathway indicates the percentage of patients experiencing the treatment. The size of the amputation cohort is visually larger here due to the data normalized at the cohort and node levels; cohort normalization adjusts sequence counts by total cohort size, while node normalization ensures flow values sum to 1 at each node, enabling visual comparison of pathway proportions as stated in the Section 2.3. Pathways colored in deep amber are the amputation pathways and those colored in pine green are non-amputation pathways.

**Figure 38 biomedicines-13-00258-f038:**
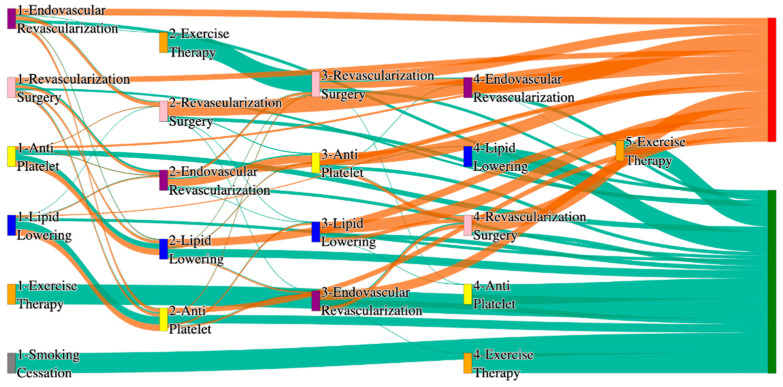
Sankey plot for the PAD patients who reportedly smoke in *All of Us* data, depicting the flow of treatments between the patients that experienced amputation and those who did not experience any amputation. Thickness of the pathway indicates the percentage of patients experiencing the treatment. The size of the amputation cohort is visually larger here due to the data normalized at the cohort and node levels; cohort normalization adjusts sequence counts by total cohort size, while node normalization ensures flow values sum to 1 at each node, enabling visual comparison of pathway proportions as stated in the Section 2.3. Pathways colored in deep amber are the amputation pathways and those colored in pine green are non-amputation pathways.

**Figure 39 biomedicines-13-00258-f039:**
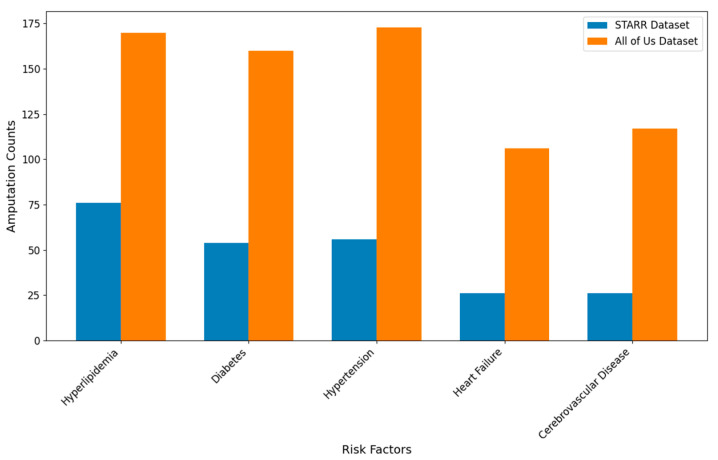
Comparison of counts of patients with amputations among various risk factors between STARR and *All of Us* datasets.

**Figure 40 biomedicines-13-00258-f040:**
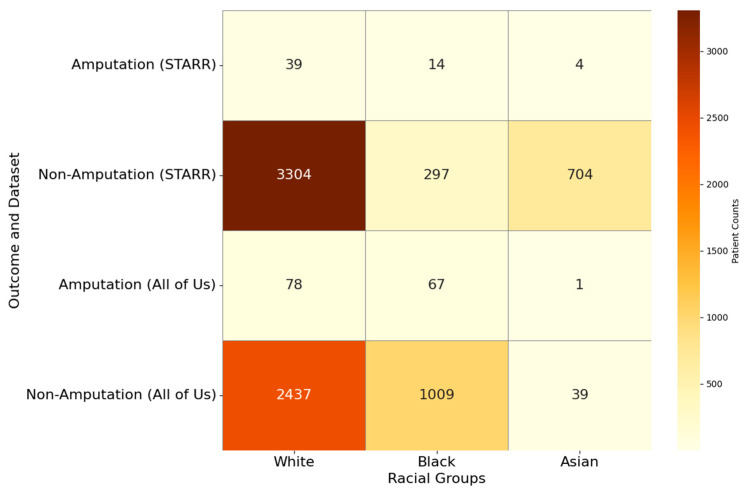
Comparison of counts of patients with amputations among various racial groups between STARR and *All of Us* datasets.

**Figure 41 biomedicines-13-00258-f041:**
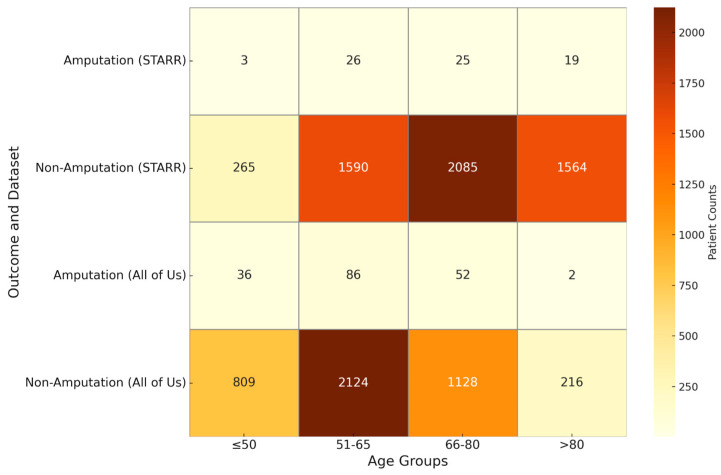
Comparison of counts of patients with and without amputations among various age groups between STARR and *All of Us* datasets.

**Table 1 biomedicines-13-00258-t001:** Counts of amputations vs. non-amputations in both the PAD datasets.

Cohort	Patients with at Least One Treatment Before Amputation	Non-Amputations
STARR Data	77	5504
*All of Us* Data	176	4085

**Table 2 biomedicines-13-00258-t002:** Pathways in the STARR dataset that exhibit higher prevalence in the amputation cohort than in the non-amputation cohort.

Amputation Pathways			
Stage 1 Treatment	Stage 2 Treatment	Stage 3 Treatment	Amputation Rate (%)
Antiplatelet	Lipid lowering	Endovascular revascularization	7.79
Antiplatelet	Revascularization surgery	Lipid lowering	5.19
Revascularization surgery	Lipid lowering	Antiplatelet	2.6
Endovascular revascularization	Lipid lowering	-	1.3

**Table 3 biomedicines-13-00258-t003:** Pathways in the STARR dataset that exhibit higher prevalence in the non-amputation cohort than in the amputation cohort.

Non-Amputation Pathways		
Stage 1 Treatment	Stage 2 Treatment	Non-Amputation Rate (%)
Antiplatelet	Lipid lowering	32
Lipid lowering	Antiplatelet	24.1
Lipid lowering	-	16.8
Antiplatelet	-	10.2

**Table 4 biomedicines-13-00258-t004:** Pathways in the *All of Us* dataset that exhibit higher prevalence in the amputation cohort than in the non-amputation cohort.

Amputation Pathways			
Stage 1 Treatment	Stage 2 Treatment	Stage 3 Treatment	Amputation Rate (%)
Endovascular revascularization	-	-	13.6
Endovascular revascularization	Revascularization surgery	-	6.25
Lipid lowering	Antiplatelet	Endovascular revascularization	3.98
Antiplatelet	Lipid lowering	Endovascular revascularization	2.84

**Table 5 biomedicines-13-00258-t005:** Pathways in the *All of Us* dataset that exhibit higher prevalence in the non-amputation cohort than in the amputation cohort.

Non-Amputation Pathways		
Stage 1 Treatment	Stage 2 Treatment	Non-Amputation Rate (%)
Lipid lowering	Antiplatelet	27.29
Antiplatelet	-	23
Antiplatelet	Lipid lowering	22
Lipid lowering	-	14

**Table 6 biomedicines-13-00258-t006:** Comparative analysis in the non-amputation cohort in both the datasets for the sequence of antiplatelet → lipid lowering.

Dataset	Treatment Sequence	Non-Amputation Rate (%)
STARR	Antiplatelet → lipid lowering	32.29
*All of Us*	Antiplatelet → lipid lowering	21.93

**Table 7 biomedicines-13-00258-t007:** Treatment sequences for amputation outcomes in both datasets.

Dataset	Treatment Sequence	Amputation Rate (%)
STARR	Antiplatelet → lipid lowering	25.97
STARR	Lipid lowering	15.58
*All of Us*	Endovascular revascularization	13.64
*All of Us*	Endovascular revascularization → revascularization surgery	6.25

**Table 8 biomedicines-13-00258-t008:** Comparative analysis of revascularization interventions in amputation cohorts of both datasets.

Dataset	Revascularization Sequence	Amputation Rate (%)
STARR	Occurrence of revascularization procedures alone	0.0
*All of Us*	Endovascular revascularization → revascularization surgery	6.25

**Table 9 biomedicines-13-00258-t009:** Peripheral overview of variations in treatment pathways in both datasets.

Dataset	Common Treatment Patterns in Amputation Cohort
STARR	Antiplatelet and lipid-lowering therapies
*All of Us*	Revascularization procedures

**Table 10 biomedicines-13-00258-t010:** Treatment pathways from odds ratio analysis in the STARR data with an odds ratio between 0 and 1, lowering the amputation risk.

Pathway	Odds Ratio
Lipid lowering, antiplatelet, revascularization surgery	0.83
Antiplatelet, lipid lowering, revascularization surgery	0.81
Lipid lowering, antiplatelet	0.70
Antiplatelet, lipid lowering	0.62
Lipid lowering, antiplatelet, endovascular revascularization	0.60

**Table 11 biomedicines-13-00258-t011:** Treatment pathways from odds ratio analysis in the *All of Us* data with an odds ratio between 0 and 1, lowering the amputation risk.

Pathway	Odds Ratio
Antiplatelet	0.35
Antiplatelet, lipid lowering	0.53
Lipid lowering	0.34

**Table 12 biomedicines-13-00258-t012:** Patient counts in STARR data—risk factors.

STARR Data with PAD		
Cohort	Patient Count with Amputation	Patient Count with Non-Amputation
STARR PAD cohort	77	5504
PAD with diabetes	54	1936
PAD with heart failure	26	1270
PAD with hypertension	56	3812
PAD with cerebrovascular disease	26	1749
PAD with coronary artery disease	43	2500
PAD with hyperlipidemia	76	4941
Patients with age > 50	74	5239
Patients with age ≤ 50	3	265
Patients with age > 65	51	3649
Patients with age ≤ 65	26	1855
Patients with age > 80	19	1564
Patients with age ≤ 80	58	4555
Male patient cohort	56	2195
Female patient cohort	21	2116
Patients who reported their race as white	39	3304
Patients who reported their race as black	14	297
Patients who reported their race as Asian	4	704
Patients who smoke	72	5143

**Table 13 biomedicines-13-00258-t013:** Patient counts in *All of Us* data—risk factors.

*All of Us* Data with PAD		
Cohort	Patient Count with Amputation	Patient Count with Non-Amputation
*All of Us* PAD cohort	176	4085
PAD with diabetes	160	2780
PAD with heart failure	106	1812
PAD with hypertension	173	3853
PAD with cerebrovascular disease	117	2454
PAD with coronary artery disease	143	2947
PAD with hyperlipidemia	170	3902
Patients with age > 50	140	3276
Patients with age ≤ 50	36	809
Patients with age > 65	54	1152
Patients with age ≤ 65	122	2933
Patients with age > 80	2	216
Patients with age ≤ 80	174	4049
Male patient cohort	105	1889
Female patient cohort	66	2063
Patients who reported their race as white	78	2437
Patients who reported their race as black	67	1009
Patients who reported their race as Asian	1	39
Patients who smoke	78	1205

## Data Availability

This study used data from the *All of Us* Research Program’s Registered Tier Dataset V7, available to authorized users on the Researcher Workbench. Given the sensitivity of the data used from the STARR, these data are not publicly available but are available to those working in direct collaboration with the corresponding author.

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
