# Peer review of "Analysis and Visualization of Confounders and Treatment Pathways Leading to Amputation and Non-Amputation in Peripheral Artery Disease Patients Using Sankey Diagrams: Enhancing Explainability"

_biomedicines, 2025, doi:10.3390/biomedicines13020258_

Round 1

Reviewer 1 Report

Comments and Suggestions for Authors

See the attached review. However, the mansucript is too long to read.

Comments on the Quality of English Language

See the attacjed revoew re[prt/ 

Author Response

1. Abstract

a. Give values.

b. Give future perspective using one line.

We sincerely thank the reviewer for their valuable comments on improving the abstract.

We have updated the abstract to include specific values, ensuring that readers can better understand the results at a glance. Additionally, we have added a future perspective in the last line, highlighting potential directions for expanding this work. Here is the updated abstract -

Background/Objectives

This study uses Sankey diagrams to analyze treatment pathways in patients with peripheral artery disease (PAD), which is a vascular condition characterized by atherosclerotic occlusion of the arteries, particularly in the lower limbs, affecting up to 14% of the general population. This study focuses on the treatment pathways that lead to amputation versus those that do not, utilizing the STARR dataset and the All of Us dataset.

Methods

The study utilized Sankey diagrams to visualize treatment pathways, highlighting the progression from initial treatments to outcomes. Odds ratio analysis was performed to quantify the association between treatment pathways and outcomes. Recognizing potential confounders, analyses were conducted by filtering patients with PAD into subgroups based on these coexisting conditions. Sankey diagrams were then generated for each sub-cohort to visualize treatment pathways.

Results

Pathways with anti-platelet and lipid-lowering treatments accounted for 56% of non-amputation cases in the STARR data and 50% in the All of Us data. Amputation pathways frequently included revascularization procedures, representing 15% of amputations in the STARR data and 20% in the All of Us data. Confounder analysis revealed that most amputated PAD patients were over 50 years old, had one or more conditions such as diabetes, hypertension, or hyperlipidemia.

Conclusions

These visualizations provide insights into treatment pathways and their associations with outcomes in PAD patients, highlighting potential impact of specific treatments on amputation and non-amputation cases. Future work builds on these findings by incorporating predictive models using machine learning techniques to further explore and quantify these relationships.

2. Introduction

a. How reliable is this visualization process to study treatment?

b. The authors should highlight the research gap and aim of the study.

We appreciate the reviewer’s constructive feedback on the introduction section.

To address the reliability of the visualization process, we have added references to a few key studies, mostly in the last paragraph of the introduction, demonstrating the utility and credibility of visualizations in understanding patterns of interests.

Furthermore, we have highlighted the research gap, emphasizing that while numerous studies focus on machine learning for PAD diagnosis, limited attention has been given to visualization-based analyses of treatment pathways. We have clarified the aim of the study to address this gap.

To address these provided comments, the papers we referred to include -

  1. Zhang, X., Padman, R., & Patel, N. (2015). Paving the COWpath: Data-Driven Approach to Clinical Pathway Analysis Using Electronic Health Record Data. Journal of Biomedical Informatics, 58, 174-186. https://doi.org/10.1016/j.jbi.2015.08.011
  2. Levy-Fixa, G., Kuperman, G. J., & Elhadad, N. (2023). Machine learning and visualization in clinical decision support: Current state and future directions. Preprint.
  3. Ross, E. G., Shah, N. H., Dalman, R. L., Nead, K. T., Cooke, J. P., & Leeper, N. J. (2016). The use of machine learning for the identification of peripheral artery disease and future mortality risk. Journal of vascular surgery, 64(5), 1515-1522.
  4. Flores, A. M., Demsas, F., Leeper, N. J., & Ross, E. G. (2021). Leveraging machine learning and artificial intelligence to improve peripheral artery disease detection, treatment, and outcomes. Circulation research, 128(12), 1833-1850.
  5. Lareyre, F., Behrendt, C. A., Chaudhuri, A., Lee, R., Carrier, M., Adam, C., ... & Raffort, J. (2023). Applications of artificial intelligence for patients with peripheral artery disease. Journal of vascular surgery, 77(2), 650-658.
  6. Albarrak, A. M. (2023). Improving the Trustworthiness of Interactive Visualization Tools for Healthcare Data through a Medical Fuzzy Expert System. Diagnostics, 13(10), 1733.

3. Methods

a. How would the combination of treatments affect the amputation procedure?

We thank the reviewer for this insightful comment.

In response, we have included a reference to a paper in the data preprocessing section discussing the impact of combination therapies such as antiplatelet and lipid-lowering treatments on amputation procedures. This strengthens the explanation of how combination therapies were factored into the study. This is one of the papers that asserts our findings in this aspect -

Belch, J. J. F., Brodmann, M., Baumgartner, I., Binder, C. J., Casula, M., Heiss, C., Kahan, T., Parini, P., Poredos, P., Catapano, A. L., & Tokgözoğlu, L. (2021). Lipid-lowering and anti-thrombotic therapy in patients with peripheral arterial disease: European Atherosclerosis Society/European Society of Vascular Medicine Joint Statement. Vasa, 50(6), 401–411. https://doi.org/10.1024/0301-1526/a000969

4. Results and Discussion

a. Data visualization should be improved.

We are grateful for the reviewer’s suggestion to enhance data visualization.

We have improved the results section by incorporating additional graphs to provide a more intuitive and engaging representation of the data alongside the Sankey diagrams. Additionally, current sankey diagrams have been updated with the color blind friendly colors.

b. How significant are these treatments?

Thank you for raising this important point.

We have validated the treatments in collaboration with a vascular surgeon, who also assisted in identifying treatment codes used to filter the data. This ensures the significance of the analyzed treatments and improves the study's reliability.

c.Is there a way to reduce the number of tables? It makes the paper too boring to read.

d. The manuscript is too long to read.

We appreciate the reviewer’s concern regarding the number of tables and the length of the manuscript.

To address this, we have moved about 70 tables from the confounder analysis to the supplementary materials, leaving the descriptions behind. This makes the manuscript more concise. While we attempted to streamline the paper, we included an additional analysis for the age group >80, <=80 as requested by reviewers, slightly increasing the manuscript length. We have ensured that the writing remains as precise as possible.

Reviewer 2 Report

Comments and Suggestions for Authors

The article uses Sankey maps to analyze and visualize the causes and treatment pathways of amputation and non-amputation in patients with peripheral artery disease.

The article can be considered for publication in "Biomedicines" after revising the following questions. The comments are below.

1) The authors' grouping of some factors is broad. Please further divide certain factors (such as age) into more intervals to explore their impact on treatment pathways and outcomes in detail and to improve the reliability of the study.

2) The authors should provide the number of samples when analyzing PAD affected by different factors to help readers understand.

3) The picture processing of this article is relatively simple. The author can add more other forms of charts to illustrate the problem intuitively when writing the article

4) In the summary analysis in Section 3.3, it is suggested that author can add graphs to visually illustrate conclusions.

Author Response

1. The authors' grouping of some factors is broad. Please further divide certain factors (such as age) into more intervals to explore their impact on treatment pathways and outcomes in detail and to improve the reliability of the study.

Thank you for your suggestion to refine factor groupings.

We have divided the age factor into an additional interval of 80 years, we already had 50 and 65 years in the paper, providing a consistent gap of 15 years between groups. This allowed us to explore treatment pathways in greater detail.

For age groups less than 50 years in the STARR dataset (amputation counts = 3) and for beyond 80 years in the All of Us dataset (amputation counts = 2), since the sample size would be insufficient for further divisions. Therefore, we confined our analysis to the age groups mentioned.

2. The authors should provide the number of samples when analyzing PAD affected by different factors to help readers understand.

We thank the reviewer for this helpful comment.

In response, we have added the number of samples for each cohort in the first sentence of every cohort description. This provides readers with a clear understanding of the patient samples analyzed for each factor.

3. The picture processing of this article is relatively simple. The authors can add more other forms of charts to illustrate the problem intuitively when writing the article.

We appreciate this valuable suggestion.

While the primary focus of the paper is on Sankey diagrams for visualizing treatment pathways, we have incorporated additional types of charts and graphs in the results section as suggested by the reviewer. These visualizations offer a more intuitive understanding of the data.

4. In the summary analysis in Section 3.3, it is suggested that the authors can add graphs to visually illustrate conclusions.

Thank you for this thoughtful comment.

In Section 3.3, we have added graphs to visually represent the summary conclusions, enhancing the readability and impact of this section.

Round 2

Reviewer 2 Report

Comments and Suggestions for Authors

The author has completed the modification according to the comments pointed out by the reviewer.